# Spine and lower body symmetry during treadmill walking in healthy individuals— In-vivo 3-dimensional kinematic analysis

**Paul Gonzalo Arauz** [1]*, **Maria-Gabriela Garcia**[1], **Patricio Chiriboga**[1‡], **Sebastian Taco-Vasquez**[2‡], **Diego Klaic**[1‡], **Emilia Verdesoto**[1‡], **Bernard Martin**[3]

**1** Colegio de Ciencias e Ingenierías "El Politécnico", Universidad San Francisco de Quito USFQ, Quito, Ecuador, **2** Departamento de Ingeniería Química, Escuela Politécnica Nacional, Quito, Ecuador, **3** Department of Industrial and Operations Engineering, University of Michigan, Ann Arbor, Michigan, United States of America

☯ These authors contributed equally to this work.
‡ PC, STV, DK and EV also contributed equally to this work.
* parauz@usfq.edu.ec

**Data Availability Statement:** Study data is publicly available at the following DOI: https://doi.org/10.5281/zenodo.7079689.

## Abstract

Although it is relevant to understand spine and lower body motions in healthy individuals for a variety of applications, such as clinical diagnosis, implant design, and the analysis of treatment outcomes, proper assessment and characterization of normative gait symmetry in healthy individuals remains unclear. The purpose of this study was to investigate the in vivo 3-dimensional (3D) spine and lower body gait symmetry kinematics during treadmill walking in healthy individuals. Sixty healthy young adults (30 males and 30 females) were evaluated during normal and fast treadmill walking using a motion capture system approach. Statistical parametric mapping and the normalized symmetry index approaches were used to determine spine, pelvis, and lower body asymmetries during treadmill walking. The spine and pelvis angular motions associated with the left and right lower limb motions, as well as the left and right lower extremity joint angles were compared for normal and fast treadmill walking. The lower lumbar left-right rotation (5.74±0.04˚) and hip internal rotation (5.33±0.18˚) presented the largest degrees of asymmetry during normal treadmill. Upper lumbar left-right lateral flexion (1.48±0.14˚) and knee flexion (2.98±0.13˚) indicated the largest asymmetries and during fast treadmill walking. Few asymmetry patterns were similar between normal and fast treadmill walking, whereas others appeared either only during normal or fast treadmill walking in this cohort of participants. These findings could provide insights into better understanding gait asymmetry in healthy individuals, and use them as reference indicators in diagnosing and evaluating abnormal gait function.

## Introduction

The spine is the central supporting structure of the torso allowing for flexibility and shock absorption, as well as routing and protecting the spinal cord. The spine has been modeled as

**Funding:** The authors received no specific funding for this work.

**Competing interests:** The authors have declared that no competing interests exist.

an inverted pendulum requiring a shifting base to maintain its stability [1–4]. Therefore, it is relevant to understand the spine and lower body motions in healthy individuals for a variety of applications, such as clinical diagnosis, implant design, and the analysis of treatment outcomes. For instance, it has been reported that low back pain may be associated with the poor movement coordination between the lumbar spine and pelvis [1, 5, 6]. Typically, individuals with low back pain present reduced lumbar spine range of motion, as well as more asymmetrical spine movement compared to healthy individuals [6–8]. In addition, lumbar total disc replacement (TDR) has been shown to provide a degree of motion preservation at the operative level during moderate speed walking [9]. Likewise, it has been reported that patients with lumbar degenerative disc disease presented an increased posterior pelvic tilt suggesting that sagittal spinopelvic deformity may predispose to anterior instability in total hip arthroplasty patients during treadmill walking [10].

Skin-marker-based motion capture systems allow for effective measurement of three-dimensional (3D) joint kinematics [11–13]. Gait symmetry has been suggested as an important indicator of gait function in impaired and healthy individuals [14–16]. Consequently, better implementation and evaluation of restorative interventions requires appropriate assessment and characterization of normative gait symmetry in healthy individuals. Several studies have investigated on gait symmetry [17–23], as well as on spine and lower body kinematics during gait [6, 8, 9, 24–27]. Despite such contributions, up to date, there is no generally accepted standard for assessing symmetry [23]. Typically, the symmetry index, symmetry ratio, and statistical approaches are implemented to determine gait symmetry [21]. Thus, it is complicated to compare among studies and establish standard criteria to guide clinical decision-making. Several studies have implemented statistical procedures to investigate interlimb asymmetries using the mean difference between left and right limbs as a symmetry parameter in pathological individuals [15, 16, 28, 29]. However, implementation of such approaches to determine reference degree of asymmetry information in healthy individuals is lacking.

Walking overground is more natural than walking on a treadmill [30–32]. However, the use of treadmills is very common in clinical and rehabilitation practices as they allow for smaller space, better control of walking speeds, and a more controlled environment for kinematics and kinetic studies [32, 33]. Currently, there is a paucity of data regarding in-vivo spine and lower body 3D kinematic asymmetries in healthy individuals during treadmill walking at different paces using statistical approaches. Therefore, the purpose of the present study was to analyze 3D spine, pelvis, and lower body gait symmetry kinematics during normal and fast treadmill walking in healthy individuals. This study applies statistical parametric mapping (SPM) [12], detecting significant differences between left and right-side movements, as well as the normalized symmetry index [21, 23] approaches to determine spine, pelvis, and lower body asymmetries. We hypothesized that there are significant differences between the spine and pelvis angular movements associated with the left lower limb motions and spine and pelvis angular movements associated with the right lower limb motions, as well as between the left and right lower extremity joint angles during normal and fast treadmill walking in healthy individuals.

## Methods

### Participants

Thirty young males and thirty young females with a healthy lifestyle (physical activity at least twice a week) participated in the present study. Exclusion criteria included: gait or cognitive impairments, disabilities, prior injuries that required surgery, auto-immune diseases, or any musculoskeletal pain. Fifty-two out of the sixty participants reported to be right-leg dominant

(with leg dominance being defined as the preferred leg for kicking a ball). On average the participant's anthropometric characteristics were 21.15 years of age (±2.47, range 18 to 30), 1.68 m (±0.1, range 1.47 to 1.88) height and 63.32 kg (±12.38, range 42 to 96) weight, and 22.32 kg/m$^2$ (±3.09, range 16.2 to 31.1) body mass index. All participants signed an informed consent form approved by the Ethics Committee of the Universidad San Francisco de Quito USFQ before data collection. This study complied with the tenets of the Declaration of Helsinki.

## Procedures and data collection

A power analysis based on our study results indicates that a sample size of 60 participants with an alpha = 0.05, and a sample ratio = 1, produces a power = 0.82. Upon arrival of the participants at the Ergonomics Laboratory of Universidad San Francisco de Quito, anthropometric data were collected. All participants first executed normal and fast level overground walking over a distance of 5 m for 4 times in separate trials. Participants were instructed to sustain a usual regular pace during normal overground walking, and accelerate their usual regular pace (as if they were in a hurry) during fast overground walking. The walking speeds of both conditions were recorded and used to set up the treadmill speeds. Participants were instructed to walk on a treadmill at normal and fast speeds under a 10-camera motion capture system (Vicon MX, Oxford, UK) surveillance sampling motion data at 100 Hz. A marker model of fifty-three reflective spherical markers (∅ 10 mm), based on previous studies [7, 12, 25], was used to obtained spine and lower body kinematics (Fig 1). The standard Vicon calibration procedures were applied to determine the 3D coordinates of the reflective spherical markers. Prior to data collection, each participant practiced for 5-minute on the treadmill. Each participant performed three trials that included at least ten complete gait cycles at normal and fast walking speeds. Thus, in total, each test condition had at least 30 complete gait cycles, and those were selected for analyses.

The spine was defined as a kinematic chain of four segments, consisting of upper thorax (T1, T6, and two midpoint markers), lower thorax (T6, L1, and two midpoint markers), upper

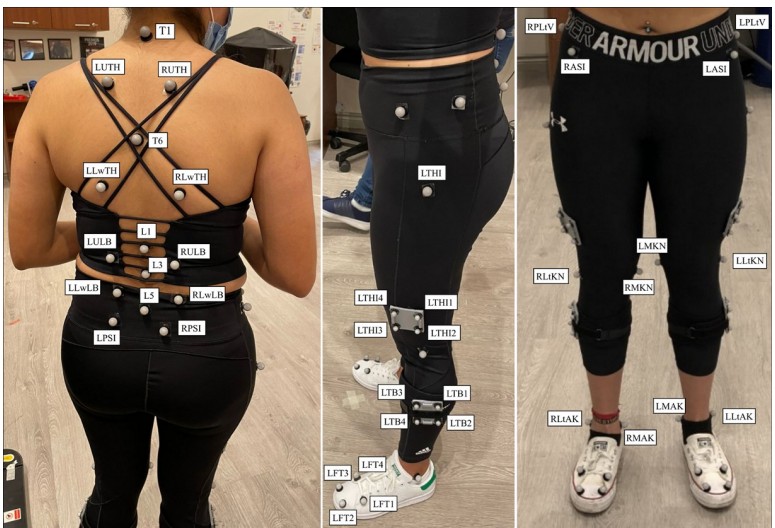

**Fig 1. Spine and lower body marker set.** Prefixes denote the following: L: Left, R: Right, U: Upper, Lw: Lower, Lt: Lateral, and M: Medial. The following landmarks were used: Spinous process at T1 (T1), spinous process at T6 (T6), spinous process at L1(L1), spinous process at L3 (L3), spinous process at L5 (L5), thorax (TH), lumbar (LB), anterior superior iliac spine (ASI), posterior superior iliac spine (PSI), femur (THI), epicondyle of femur (KN), tibia (TB), malleoli (AK), and foot (FT).

lumbar (L1, L3, and two midpoint markers), and the lower lumbar (L3, L5, and two midpoint markers) segments [7] (Fig 2). Local $z$ axes were determined on spine segments between T1 and T6, T6 and L1, L1 and L3, and L3 and L5 for the upper thorax, lower thorax, upper lumbar, and lower lumbar segments, respectively. The local $x$ axis, pointing anteriorly, of each spine segment was calculated by the cross product between the local $z$ axis and the vector defined by the two midpoint markers (Fig 2). 3D angles of the upper thorax (between lower thorax and upper thorax), lower thorax (between upper lumbar and lower thorax), upper lumbar (between lower lumbar and upper lumbar), and lower lumbar (between pelvis and lower lumbar) were calculated (Fig 2). The cross product of opposite anterior superior iliac spine (ASIS) and posterior superior iliac spine (PSIS) left and right markers defined the local pelvis $z$ axis, pointing upwards. The $y$ axis, pointing laterally, was defined between left and right ASIS (Fig 2). Pelvis rotations were calculated relative to the global Vicon coordinate system (Fig 2). The long axes of the femur, tibia, and foot determined local $z$ axes of each segment, respectively. Lateral and medial markers on the knee, ankle, and foot determined the local $y$ axes, pointing laterally, of the femur, tibia, and foot, respectively [12, 34]. Those axes were projected on thigh, tibia, and foot clusters and determined the 3D joint angles of the hip (between pelvis and femur), knee (between femur and tibia), and ankle (between foot and tibia) (Fig 2). Segment and joint angles were calculated using a Cardan angle sequence [35] (Fig 2). Data were exported and processed in MATLAB (MathWorks, Inc., Natick, MA) using a custom program.

The spine and pelvis 3D angular motions associated with the left and right lower limb movements, as well as the left and right lower extremity angular movements were compared for treadmill walking. Segment and joint angles for a standing relaxed pose were utilized as the zero neutral reference (Fig 2). The angular data was split into individual strides, and a time normalized waveform (0–100%) of the average gait cycle was generated with 1% sample steps [12, 15, 16], with 0% corresponding to heel contact of the concerned leg. Strides were defined to start with the initial contact and end with the following initial contact of one foot [23]. The 3D angles of the upper thorax, lower thorax, upper lumbar, lower lumbar and pelvis segments, as well as the hip, knee, and ankle joints were calculated to evaluate spine, pelvis and lower body kinematic gait symmetry.

Symmetry was calculated throughout the gait cycle for spine, pelvis, and lower body motions. Statistical parametric mapping and the normalized symmetry index, presented by Gouwanda et al. [21], were calculated for assessing gait symmetry of the spine and pelvis angular motions, as well as lower body joint angles. The normalized symmetry index ($SI_{norm}$) was calculated based on Eq 1 [19–21, 23].

$$SI_{norm} = \frac{X_{norm(R)} - X_{norm(L)}}{0.5 * (X_{norm(R)} + X_{norm(L)})} * 100\%, \quad \text{with } X_{norm(n)} = \frac{X_n - X_{min}}{X_{max} - X_{min}} + 1 \tag{1}$$

## Statistical analysis

The software MATLAB (MathWorks, Inc., Natick, MA) was used to performed SPM [12, 36–38] analyses using scalar fields to determine significant differences between the spine and pelvis angular motions associated with the left lower limb motions, and spine and pelvis angular motions associated with the right lower limb motions, as well as between the left and right hip, knee, and ankles joint angles throughout the gait cycle. A Student's t-test was used to compare maximum $SI_{norm}$ differences between normal and fast treadmill walking. Likewise, A Student's t-test compared walking speeds for each condition. A significance level of $\alpha = 0.05$ was used for the analysis.

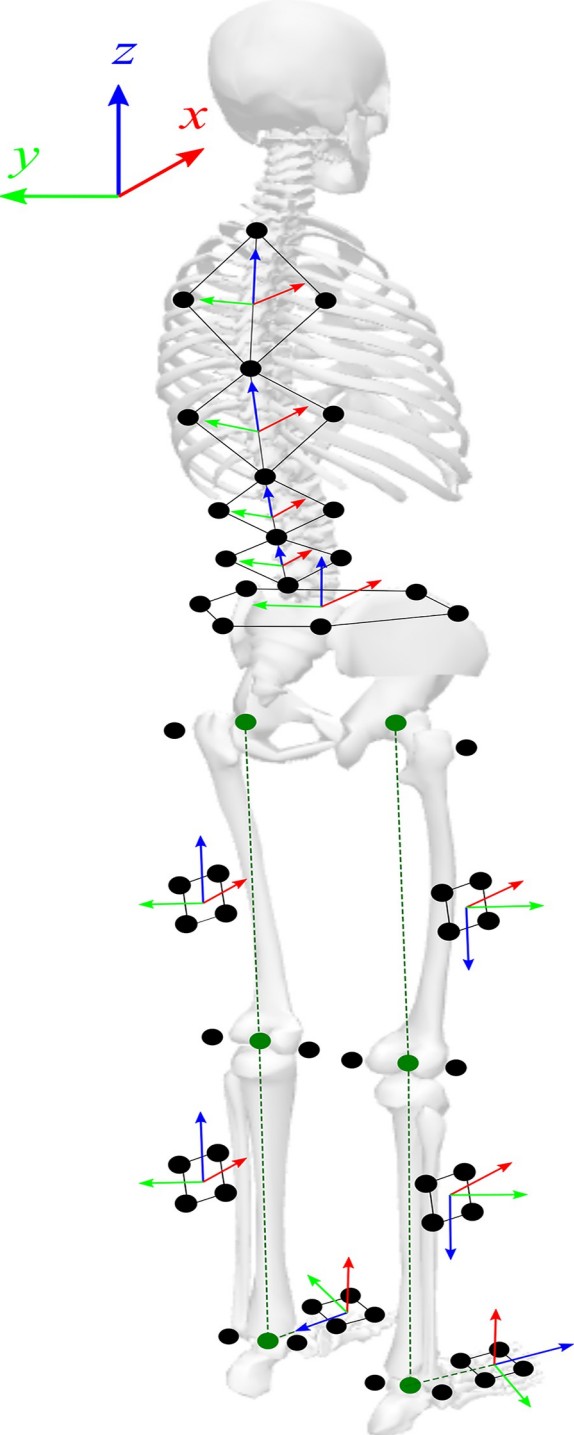

**Fig 2. Three-dimensional coordinate systems defined for upper thorax, lower thorax, upper lumbar, lower lumbar, pelvis, left and right thigh, left and right tibia, and left and right foot segments.** Local z axes were determined between T1 and T6, T6 and L1, L1 and L3, and L3 and L5 for upper thorax, lower thorax, upper lumbar, and lower lumbar segments, respectively. Cross product of the z axis and the vector defined by the two midpoint markers determined the x axis of each spine segment. Joint angles defined for the upper thorax, lower thorax, upper lumbar, and lower lumbar. The left and right anterior superior iliac spine (ASIS) and posterior superior iliac spine (PSIS) markers defined the local pelvis axes, with the y axis defined between left and right ASIS, and the x axis pointing anteriorly. Anatomical hip, knee, ankle joint axes were projected on thigh, tibia, and foot clusters, respectively, with the local z axis along the long axis of the femur, tibia, and foot, and the local y axis pointing laterally.

## Results

### Walking speed

Participants walked with an average normal treadmill walking speed of 1.19 m/s (±0.15, range 0.84 to 1.64). This was significantly smaller ($p<0.001$) than the average fast treadmill walking speed 1.79 m/s (±0.2 range 1.43 to 2.31).

### Asymmetric spine motion during normal treadmill walking

SPM analysis indicated that the upper and lower thorax segments presented symmetrical angular motions during normal treadmill walking, as the scalar field SPM curve did not exceed the threshold t* for α = 0.05 (Fig 3). $SI_{norm}$ values for upper and lower thorax flexion-extension varied between ±35% whereas the left-right lateral flexion and left-right rotation varied between ±15% (Fig 3). SPM indicated that upper and lower lumbar angular motions were asymmetrical. The upper lumbar indicated an asymmetrical flexion-extension at 45–52% and 93–99% of the gait cycle, with $SI_{norm}$ values varying between ±35% (Fig 3). The upper lumbar left-right lateral flexion was asymmetrical throughout the normal treadmill walking cycle, with $SI_{norm}$ values changing between ±15% (Fig 3). No asymmetrical motion was detected for the upper lumbar left-right rotation, and the $SI_{norm}$ values varied between ±12% (Fig 3). The lower lumbar flexion-extension was asymmetrical at 2–3%, 4–16%, and 56–66% of the gait cycle, with the $SI_{norm}$ values varying between ±28% (Fig 3). The lower lumbar left-right lateral flexion was asymmetrical at 1–8%, 14–45%, 60–98%, and 99–100% of the gait cycle, with the $SI_{norm}$ values varying between ±14% (Fig 3). The lower lumbar left-right rotation was asymmetrical during the complete gait cycle, with the $SI_{norm}$ values changing between ±13% (Fig 3).

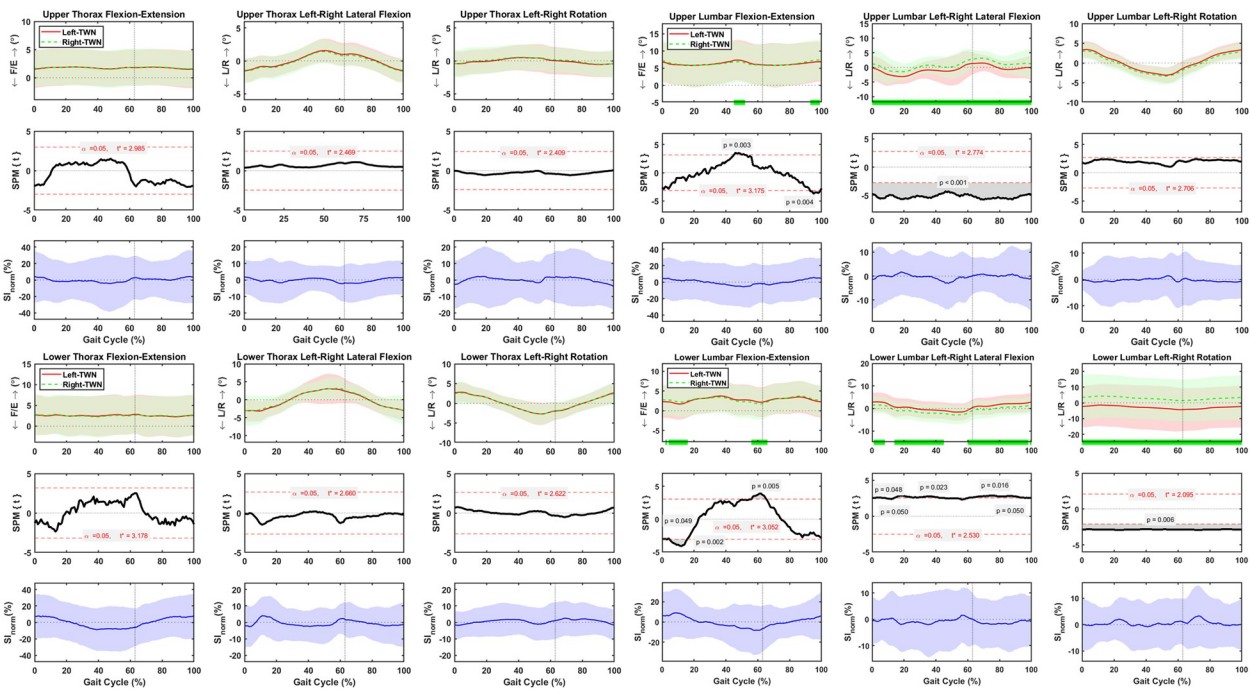

**Fig 3. Average and standard deviation of upper thorax, lower thorax, upper lumbar, and lower lumbar flexion-extension (F/E), left-right (L/R) lateral flexion, and (L/R) rotation, for left and right sides during one gait cycle of normal treadmill walking (TWN) in sixty healthy participants.** Green bars on the horizontal axis and the scalar field SPM results with threshold t* depict where, in % cycle, left side angles were greater or lesser than right side angles. The normalized symmetry index ($SI_{norm}$) calculated during one gait cycle of TWN. Solid and dashed lines correspond to average left and right sides, as well as average $SI_{norm}$, and shaded areas correspond to standard deviation. Black dotted vertical lines denote toe-off.

## Asymmetric spine motions during fast treadmill walking

SPM analysis showed that the upper and lower thorax segments presented symmetrical angular motions during fast treadmill walking (Fig 4). $SI_{norm}$ values for upper and lower thorax flexion-extension varied between ±36% whereas the left-right lateral flexion and left-right rotation varied between ±17% (Fig 4). SPM indicated that upper and lower lumbar angular motions were asymmetrical. The upper lumbar indicated a symmetricalflexion-extension, with $SI_{norm}$ values varying between ±25% (Fig 4). The upper lumbar left-right lateral flexion was asymmetrical throughout the fast treadmill walking cycle, with $SI_{norm}$ values changing between ±14% (Fig 4). The upper lumbar presented an asymmetrical left-right rotation at 13–20% and 60–73% of the gait cycle, with the $SI_{norm}$ values varying between ±12% (Fig 4). The lower lumbar flexion-extension was asymmetrical at 9–19% and 58–68% of the gait cycle, with the $SI_{norm}$ values varying between ±29% (Fig 4). The lower lumbar left-right lateral flexion was symmetrical, and the $SI_{norm}$ values varied between ±14% (Fig 4). The lower lumbar left-right rotation was symmetrical, and the $SI_{norm}$ values changed between ±14% (Fig 4).

Descriptive statistics of the average degree of asymmetry, describing the mean difference between left and right-side movements when the scalar field SPM detected significant differences, and the maximum magnitude of the $SI_{norm}$ when the scalar field SPM detected significant differences, in spine segments during normal and fast treadmill walking are presented in Table 1.

## Asymmetric lower body motions during normal treadmill walking

Pelvis posterior-anterior tilt was asymmetrical at 30–40%, 48–49%, and 76–93% of the gait cycle, with the $SI_{norm}$ values changing between ±25% (Fig 5). Pelvis left-rightobliquity was

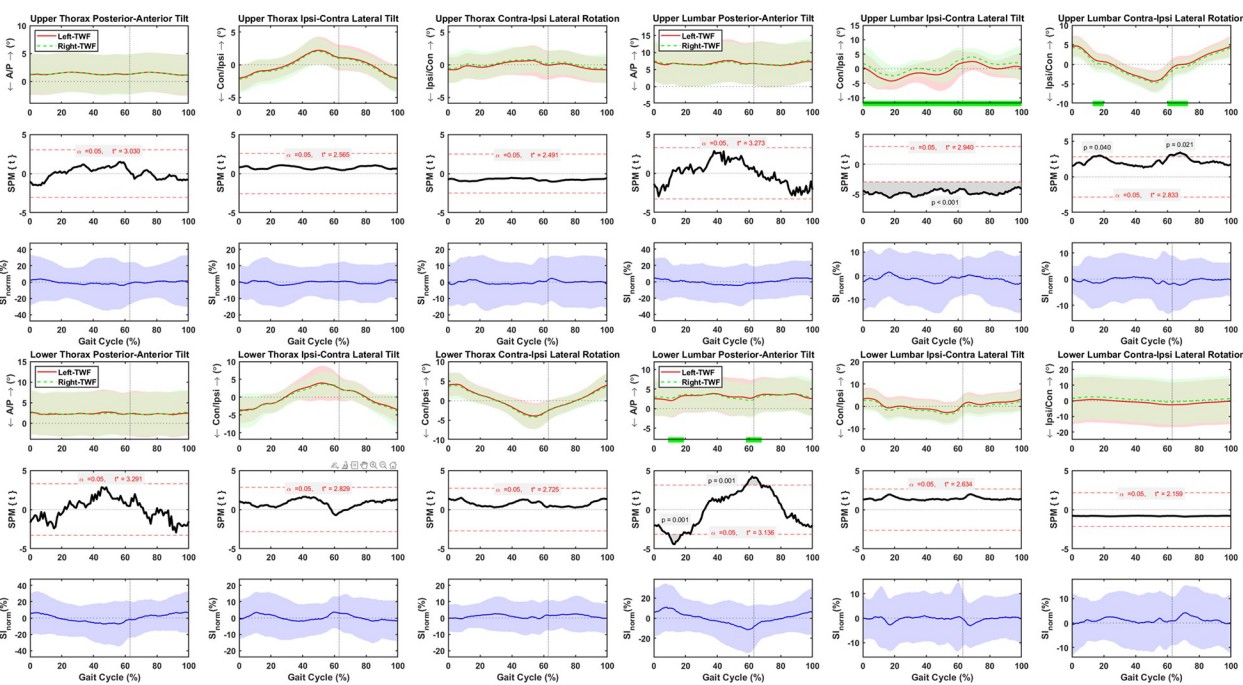

**Fig 4. Average and standard deviation of upper thorax, lower thorax, upper lumbar, and lower lumbar flexion-extension (F/E), left-right (L/R) lateral flexion, and (L/R) rotation, for left and right sides during one gait cycle of fast treadmill walking (TWF) in sixty healthy participants.** Green bars on the horizontal axis and the scalar field SPM results with threshold t* depict where, in % cycle, left side angles were greater or lesser than right side angles. The normalized symmetry index ($SI_{norm}$) calculated during one gait cycle of TWF. Solid and dashed lines correspond to average left and right sides, as well as average $SI_{norm}$, and shaded areas correspond to standard deviation. Black dotted vertical lines denote toe-off.

**Table 1. Average and standard deviation of the degree of asymmetry and maximum magnitude of the normalized symmetry index between the associated left and right motions in spine segments during normal and fast treadmill walking of N = 60 participants.**

| Segment | Treadmill Walking Normal | | | | Treadmill Walking Fast | | | |
|---|---|---|---|---|---|---|---|---|
| *Upper Lumbar* | *Flexion-Extension* | | | | *Flexion-Extension* | | | |
| | *Gait Cycle %* | *DoA* | *p-value* | *max SI$_{norm}$ %* | *Gait Cycle %* | *DoA* | *p-value* | *max SI$_{norm}$ %* |
| | 45 to 52 | 0.46±0.03 | 0.003 | 30 | | | | |
| | 93 to 99 | 0.48±0.05 | 0.004 | 30 | | | | |
| | *Left-Right Lateral Flexion* | | | | *Left-Right Lateral Flexion* | | | |
| | *Gait Cycle %* | *DoA* | *p-value* | *max SI$_{norm}$ %* | *Gait Cycle %* | *DoA* | *p-value* | *max SI$_{norm}$ %* |
| | 0 to 100 | 1.54±0.11 | <0.001 | 13 | 0 to 100 | 1.48±0.14 | <0.001 | 15.72 |
| | *Left-Right Rotation* | | | | *Left-Right Rotation* | | | |
| | Gait Cycle % | *DoA* | *p-value* | *max SI$_{norm}$ %* | *Gait Cycle %* | *DoA* | *p-value* | *max SI$_{norm}$ %* |
| | | | | | 13 to 20 | 0.64±0.016 | 0.04 | 11 |
| | | | | | 60 to 73 | 0.68±0.054 | 0.021 | 13.3 |
| *Lower Lumbar* | *Flexion-Extension* | | | | *Flexion-Extension* | | | |
| | *Gait Cycle %* | *DoA* | *p-value* | *max SI$_{norm}$ %* | *Gait Cycle %* | *DoA* | *p-value* | *max SI$_{norm}$ %* |
| | 2 to 3 | 0.48±0.01 | 0.049 | 29 | 9 to 19 | 0.68±0.05 | 0.001 | 30 |
| | 4 to 16 | 0.45±0.09 | 0.002 | 32.7 | 58 to 68 | 0.51±0.12 | 0.001 | 34 |
| | 55 to 66 | 0.42±0.09 | 0.005 | 32.7 | | | | |
| | *Left-Right Lateral Flexion* | | | | *Left-Right Lateral Flexion* | | | |
| | *Gait Cycle %* | *DoA* | *p-value* | *max SI$_{norm}$ %* | *Gait Cycle %* | *DoA* | *p-value* | *max SI$_{norm}$ %* |
| | 1 to 8 | 1.3±0.02 | 0.048 | 9.7 | | | | |
| | 14 to 45 | 1.42±0.06 | 0.023 | 14 | | | | |
| | 60 to 98 | 1.42±0.08 | 0.016 | 12.5 | | | | |
| | 99 to 100 | 1.32±0.03 | 0.05 | 10 | | | | |
| | *Left-Right Rotation* | | | | *Left-Right Rotation* | | | |
| | *Gait Cycle %* | *DoA* | *p-value* | *max SI$_{norm}$ %* | *Gait Cycle %* | *DoA* | *p-value* | *max SI$_{norm}$ %* |
| | 0 to 100 | 5.74±0.04 | 0.006 | 14.73 | | | | |

Abbreviations: DoA, degree of asymmetry; SI$_{norm}$, normalized symmetry index.

asymmetrical throughout the normal treadmill walking cycle, with SI$_{norm}$ values changing between ±12% (Fig 5). No asymmetries were detected for pelvis left-right rotations, and the SI$_{norm}$ values varied between ±11% (Fig 5). Significant flexion-extension asymmetries were detected between left and right hips at 15–55% of the normal treadmill walking cycle, with SI$_{norm}$ values varying between ±12% (Fig 5). Adduction-abduction of left and right hips were symmetrical, and the SI$_{norm}$ values varied between ±18% (Fig 5). Hip internal-external rotation was asymmetrical at 4–10% and 68–78% of the gait cycle, with SI$_{norm}$ values varying between ±23% (Fig 5). Right knees were more flexed than the left ones at 5–39% and 82–97% of the gait cycle, with SI$_{norm}$ values varying between ±8% (Fig 5). No asymmetries were detected for knee adduction-abduction, and the SI$_{norm}$ values varied between ±22% (Fig 5). Left knees had less internal rotation than the right knees at 4–10%, 16–17%, 22–42%, 83–87%, and 96–97% of the gait cycle, with SI$_{norm}$ values varying between ±20% (Fig 5). Neither ankle dorsi-plantar flexion nor internal-external rotation were asymmetrical during normal treadmill walking, and the SI$_{norm}$ values varied between ±10% and ±29%, respectively (Fig 5). Yet, the right ankles had more eversion than the left ones at 0–19% and 71–100% of the gait cycle, with SI$_{norm}$ values varying between ±16% (Fig 5). The standard deviation of the left side was higher than the right side for hip, knee, and ankle motions (Fig 5).

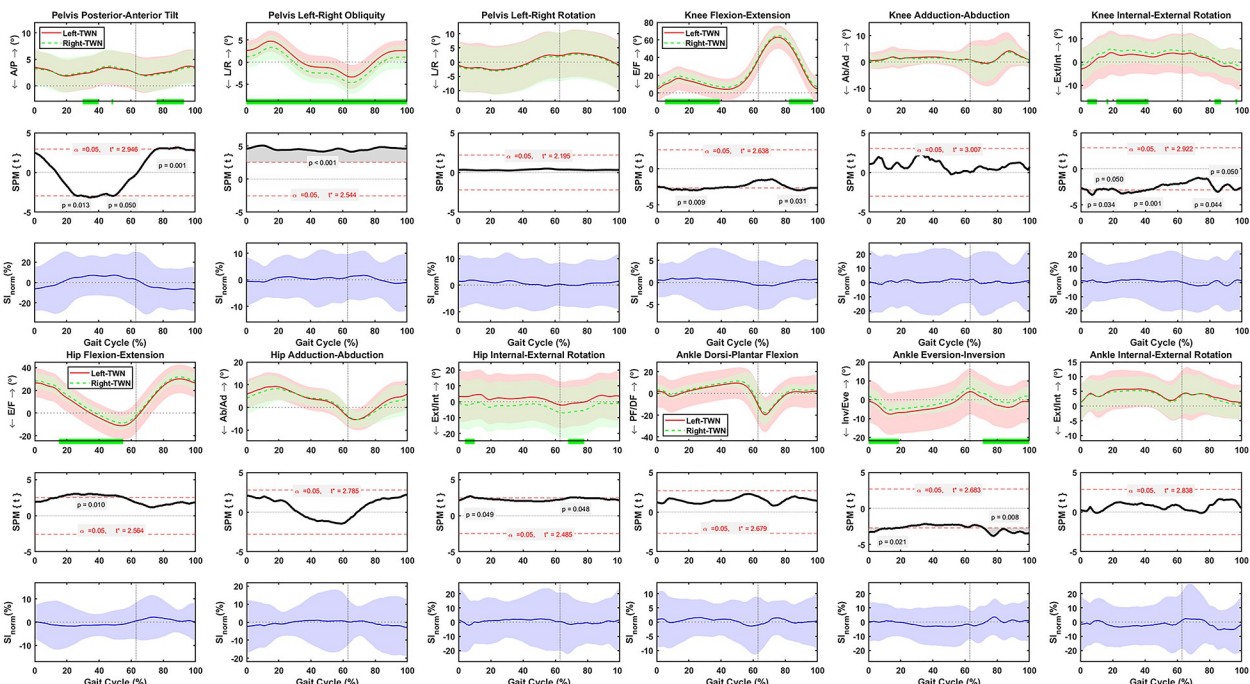

**Fig 5. Average and standard deviation of pelvis posterior-anterior (P/A) tilt, left-right(L/R) obliquity, and (L/R) rotation, hip and knee flexion-extension (F/E), adduction-abduction (Ad/Ab), and internal-external (Int/Ext) rotation, and ankle dorsi-plantar flexion (DF/PF), eversion-inversion (Eve/Inv), and internal-external (Int/Ext) rotation for left and right sides during one gait cycle of normal treadmill walking (TWN) in sixty healthy participants.** Green bars on the horizontal axis and the scalar field SPM results with threshold t* t depict where, in % cycle, left side angles were greater or lesser than right side angles. The normalized symmetry index (SI_norm) calculated during one gait cycle of TWN. Solid and dashed lines correspond to average left and right sides, as well as average SI_norm, and shaded areas correspond to standard deviation. Black dotted vertical lines denote toe-off.

### Asymmetric lower body motions during fast treadmill walking

Pelvis posterior-anterior tilt was asymmetrical at 0–13%, 36–63%, and 84–100% of the gait cycle, with the SI_norm values changing between ±29% (Fig 6). Pelvis left-right obliquity was asymmetrical at 4–22% and 53–66% of the gait cycle, with the SI_norm values changing between ±11% (Fig 6). No asymmetries were detected for pelvis left-right rotations, and the SI_norm values varied between ±11% (Fig 6). Significant flexion-extension asymmetries were detected between left and right hips at 19–53% of the fast treadmill walking cycle, with SI_norm values varying between ±10% (Fig 6). Adduction-abduction of left and right hips were symmetrical, and the SI_norm values varied between ±19% (Fig 6). Hip internal-external rotation was symmetrical, and SI_norm values varied between ±22% (Fig 6). Right knees were more flexed than the left ones at 0–42% and 81–100% of the gait cycle, with SI_norm values varying between ±9% (Fig 6). No asymmetries were detected for knee adduction-abduction, and the SI_norm values varied between ±21% (Fig 6). Left knees had less internal rotation than the right knees at 0–20%, 25–40%, and 95–100% of the gait cycle, with SI_norm values varying between ±21% (Fig 6). Ankle dorsi-plantar flexion was asymmetrical at 12–13%,35–37%, 45–48% and 79–94% of the gait cycle, with SI_norm values varying between ±12% (Fig 6). Ankle eversion-inversion was symmetrical, and the SI_norm values varied between ±15% (Fig 6). Ankle internal-external rotation was asymmetrical at 92–98% of the gait cycle, with the SI_norm values changing between ±22% (Fig 6). The standard deviation of the left side was higher than the right side for hip, knee, and ankle motions (Fig 6).

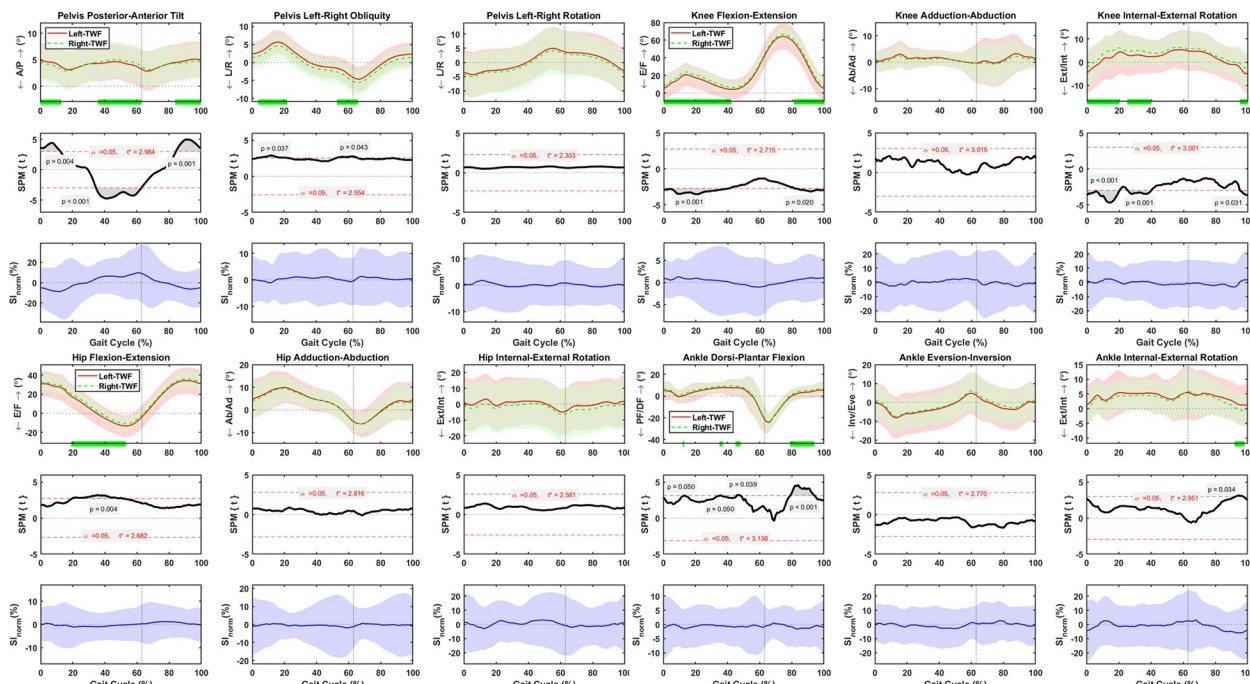

**Fig 6. Average and standard deviation of pelvis posterior-anterior (P/A) tilt, left-right (L/R) obliquity, and (L/R) rotation, hip and knee flexion-extension (F/E), adduction-abduction (Ad/Ab), and internal-external (Int/Ext) rotation, and ankle dorsi-plantar flexion (DF/PF), eversion-inversion (Eve/Inv), and internal-external (Int/Ext) rotation for left and right sides during one gait cycle of normal treadmill walking (TWN) in sixty healthy participants.** Green bars on the horizontal axis and the scalar field SPM results with threshold t* t depict where, in % cycle, left side angles were greater or lesser than right side angles. The normalized symmetry index (SI_norm) calculated during one gait cycle of TWN. Solid and dashed lines correspond to average left and right sides, as well as average SI_norm, and shaded areas correspond to standard deviation. Black dotted vertical lines denote toe-off.

Descriptive statistics of the average degree of asymmetry, describing the mean difference between left and right-side movements when the scalar field SPM detected significant differences, and the maximum magnitude of the SI_norm when the scalar field SPM detected significant differences, in the pelvis segment and lower body joints during normal and fast treadmill walking are presented in Table 2.

Descriptive statistics of the SI_norm and its comparison between normal and fast treadmill walking is presented in Table 3. Overall, greater asymmetries were found during fast treadmill walking than normal treadmill walking.

## Discussion

The purpose of the present study was to examine 3D spine, pelvis, and lower body symmetry kinematics in young healthy individuals throughout the gait cycle during normal and fast treadmill walking. Our analysis revealed significant asymmetries in upper lumbar, lower lumbar, and pelvis segments, as well as in hip, knee, and ankle joints during normal and fast treadmill walking. Degrees of asymmetry and the associated maximum magnitude of SI_norm of 5.74 ±0.04˚ and 14%, as well as 5.33±0.18˚ and 21%, for the lower lumbar left-right rotation and hip internal rotation, respectively, were the largest asymmetries detected during normal treadmill walking. Upper lumbar left-right lateral flexion and knee flexion-extension with degrees of asymmetry and the associated the maximum magnitude of SI_norm of 1.48±0.14˚ and 15.3%, as well as 2.98±0.13˚ and 6.5%, respectively, were the largest asymmetries found during fast treadmill walking. The current analysis revealed that few asymmetry patterns were similar

**Table 2. Average and standard deviation of the degree of asymmetry and maximum magnitude of the normalized symmetry index between the associated left and right motions in the pelvis segment and the lower body joints during normal and fast treadmill walking of N = 60 participants.**

| Segment | Treadmill Walking Normal | | | | Treadmill Walking Fast | | | |
|---|---|---|---|---|---|---|---|---|
| Pelvis | *Posterior-Anterior Tilt* | | | | *Posterior-Anterior Tilt* | | | |
| | *Gait Cycle %* | *DoA* | *p-value* | *max SI$_{norm}$ %* | *Gait Cycle %* | *DoA* | *p-value* | *max SI$_{norm}$ %* |
| | 30 to 40 | 0.22±0.01 | 0.013 | 26.8 | 0 to 13 | 051±0.03 | 0.004 | 35.8 |
| | 48 to 49 | 0.26±0.01 | 0.05 | 27.6 | 36 to 63 | 0.46±0.03 | <0.001 | 37.4 |
| | 76 to 93 | 0.22±0.02 | 0.001 | 27.2 | 84 to 100 | 0.48±0.05 | 0.001 | 24.6 |
| | *Left-Right Obliquity* | | | | *Left-Right Obliquity* | | | |
| | *Gait Cycle %* | *DoA* | *p-value* | *max SI$_{norm}$ %* | *Gait Cycle %* | *DoA* | *p-value* | *max SI$_{norm}$ %* |
| | 0 to 100 | 1.38±0.06 | 0.001 | 11.5 | 4 to 22 | 1.06±0.06˚ | 0.037 | 10.3 |
| | | | | | 53 to 66 | 1.07±0.03˚ | 0.043 | 11 |
| Hip | *Flexion-Extension* | | | | *Flexion-Extension* | | | |
| | *Gait Cycle %* | *DoA* | *p-value* | *max SI$_{norm}$ %* | *Gait Cycle %* | *DoA* | *p-value* | *max SI$_{norm}$ %* |
| | 15 to 55 | 2.57±0.08 | 0.01 | 11.8 | 19 to 53 | 2.60±0.13 | 0.004 | 10 |
| | *Internal-External Rotation* | | | | *Internal-External Rotation* | | | |
| | *Gait Cycle %* | *DoA* | *p-value* | *max SI$_{norm}$ %* | *Gait Cycle %* | *DoA* | *p-value* | *max SI$_{norm}$ %* |
| | 4 to 10 | 5.33±0.18 | 0.049 | 21 | | | | |
| | 68 to 78 | 5.27±0.08 | 0.048 | 22 | | | | |
| Knee | *Flexion-Extension* | | | | *Flexion-Extension* | | | |
| | *Gait Cycle %* | *DoA* | *p-value* | *max SI$_{norm}$ %* | *Gait Cycle %* | *DoA* | *p-value* | *max SI$_{norm}$ %* |
| | 5 to 39 | 2.65±0.1 | 0.009 | 7.9 | 0 to 42 | 2.81±0.24 | 0.001 | 8.3 |
| | 82 to 97 | 2.71±0.13 | 0.031 | 5.3 | 81 to 100 | 2.98±0.13 | 0.02 | 6.5 |
| | *Internal-External Rotation* | | | | *Internal-External Rotation* | | | |
| | *Gait Cycle %* | *DoA* | *p-value* | *max SI$_{norm}$ %* | *Gait Cycle %* | *DoA* | *p-value* | *max SI$_{norm}$ %* |
| | 4 to 10 | 1.85±0.12 | 0.049 | 18.5 | 0 to 20 | 2.35±0.35 | <0.001 | 22.7 |
| | 16 to 17 | 1.64±0.01 | 0.05 | 12.6 | 25 to 40 | 1.89±0.11 | 0.001 | 15.4 |
| | 22 to 42 | 1.75±0.1 | 0.001 | 18.5 | 95 to 100 | 2.29±0.25 | 0.031 | 22.7 |
| | 83 to 87 | 1.57±0.08 | 0.044 | 18.5 | | | | |
| | 96 to 97 | 1.75±0.1 | 0.05 | 22.7 | | | | |
| Ankle | *Dorsi-Plantar Flexion* | | | | *Dorsi-Plantar Flexion* | | | |
| | *Gait Cycle %* | *DoA* | *p-value* | *max SI$_{norm}$ %* | *Gait Cycle %* | *DoA* | *p-value* | *max SI$_{norm}$ %* |
| | | | | | 12 to 13 | 1.62±0.08 | 0.05 | 8.4 |
| | | | | | 35 to 37 | 1.21±0.01 | 0.05 | 6.5 |
| | | | | | 45 to 48 | 1.34±0.16 | 0.039 | 7.8 |
| | | | | | 79 to 94 | 1.37±0.14 | <0.001 | 8.4 |
| | *Eversion-Inversion* | | | | *Eversion-Inversion* | | | |
| | *Gait Cycle %* | *DoA* | *p-value* | *max SI$_{norm}$ %* | *Gait Cycle %* | *DoA* | *p-value* | *max SI$_{norm}$ %* |
| | 0 to 19 | 2.34±0.18 | 0.021 | 13.4 | | | | |
| | 71 to 100 | 2.6±0.22 | 0.008 | 16.4 | | | | |
| | *Internal-External Rotation* | | | | *Internal-External Rotation* | | | |
| | *Gait Cycle %* | *DoA* | *p-value* | *max SI$_{norm}$ %* | *Gait Cycle %* | *DoA* | *p-value* | *max SI$_{norm}$ %* |
| | | | | | 92 to 98 | 1.76±0.05 | 0.034 | 24.5 |

Abbreviations: DoA, degree of asymmetry; SI$_{norm}$, normalized symmetry index.

between normal and fast treadmill walking, whereas others appeared either only during normal or fast treadmill walking. These results rejected the null hypothesis of no difference in spine, pelvis, and lower body motions between left and right sides during normal and fast treadmill walking in healthy individuals.

**Table 3. Descriptive statistics of the maximum $SI_{norm}$ in % and its comparison between normal and fast treadmill walking for N = 60 participants.**

| Segment | Motion | Treadmill Walking Normal | | | | Treadmill Walking Fast | | | | |
|---|---|---|---|---|---|---|---|---|---|---|
| | | *Mean* | *SD* | *Max* | *Min* | *Mean* | *SD* | *Max* | *Min* | *p-value* |
| *Upper Thorax* | *Flexion-Extension* | 45.56 | 12.47 | 66.67 | 12.70 | 44.17 | 12.61 | 65.61 | 12.64 | 0.201 |
| | *Left-Right Lateral Flexion* | 13.87 | 11.07 | 60.71 | 0.56 | 15.36 | 12.07 | 63.82 | 0.84 | 0.109 |
| | *Left-Right Rotation* | 22.28 | 14.08 | 64.70 | 0.81 | 22.59 | 12.30 | 58.93 | 1.28 | 0.765 |
| *Lower Thorax* | *Flexion-Extension* | 44.88 | 11.76 | 66.50 | 15.60 | 42.42 | 11.28 | 65.84 | 9.50 | **0.019** |
| | *Left-Right Lateral Flexion* | 15.11 | 11.30 | 56.47 | 0.46 | 17.82 | 12.17 | 59.03 | 0.00 | **0.000** |
| | *Left-Right Rotation* | 13.60 | 9.13 | 60.25 | 2.03 | 13.69 | 10.75 | 66.07 | 0.54 | 0.919 |
| *Upper Lumbar* | *Flexion-Extension* | 42.05 | 11.84 | 66.67 | 9.69 | 41.35 | 11.09 | 66.67 | 16.63 | 0.422 |
| | *Left-Right Lateral Flexion* | 13.77 | 10.86 | 64.38 | 0.00 | 13.65 | 10.10 | 59.13 | 0.18 | 0.867 |
| | *Left-Right Rotation* | 10.87 | 8.21 | 64.26 | 1.05 | 11.42 | 9.24 | 66.67 | 0.41 | 0.456 |
| *Lower Lumbar* | *Flexion-Extension* | 37.29 | 12.85 | 66.67 | 11.15 | 37.79 | 12.95 | 66.67 | 12.54 | 0.574 |
| | *Left-Right Lateral Flexion* | 14.82 | 10.16 | 55.32 | 0.00 | 15.90 | 11.83 | 59.77 | 0.59 | 0.163 |
| | *Left-Right Rotation* | 14.51 | 9.99 | 56.92 | 0.05 | 16.82 | 11.77 | 63.86 | 0.75 | **0.002** |
| *Pelvis* | *Posterior-Anterior Tilt* | 38.10 | 14.02 | 66.67 | 6.40 | 37.77 | 13.83 | 66.67 | 4.94 | 0.770 |
| | *Left-Right Obliquity* | 9.83 | 10.20 | 65.57 | 0.28 | 10.49 | 11.56 | 62.98 | 0.05 | 0.474 |
| | *Left-Right Rotation* | 12.25 | 8.51 | 50.01 | 0.67 | 11.88 | 8.95 | 59.82 | 0.29 | 0.602 |
| *Hip* | *Flexion-Extension* | 7.98 | 10.13 | 65.29 | 0.15 | 8.18 | 8.86 | 60.56 | 0.33 | 0.787 |
| | *Adduction-Abduction* | 15.39 | 15.10 | 66.67 | 0.84 | 16.40 | 14.69 | 66.62 | 1.59 | 0.372 |
| | *Internal-External Rotation* | 28.46 | 15.05 | 66.06 | 3.25 | 29.51 | 13.59 | 63.38 | 5.99 | 0.164 |
| *Knee* | *Flexion-Extension* | 6.62 | 6.64 | 61.39 | 0.63 | 7.71 | 6.19 | 62.62 | 0.40 | **0.045** |
| | *Adduction-Abduction* | 29.80 | 16.00 | 66.05 | 2.38 | 30.33 | 13.42 | 63.55 | 2.61 | 0.545 |
| | *Internal-External Rotation* | 30.06 | 13.17 | 62.85 | 6.59 | 27.71 | 12.44 | 62.08 | 4.34 | **0.009** |
| *Ankle* | *Dorsi-Plantar Flexion* | 11.77 | 9.08 | 66.67 | 0.64 | 10.91 | 7.88 | 58.57 | 1.045 | 0.294 |
| | *Eversion-Inversion* | 21.22 | 10.39 | 57.16 | 1.87 | 22.97 | 11.76 | 63.11 | 1.98 | **0.036** |
| | *Internal-External Rotation* | 30.00 | 14.85 | 66.61 | 1.17 | 31.13 | 14.18 | 66.67 | 1.22 | 0.261 |

Abbreviations: SD, standard deviation; Max, maximum; Min, minimum.

It has been reported that the walking speed affects individuals' gait kinematics [39, 40]; however, the influence of treadmill walking speed on gait symmetry kinematics in healthy individuals remains unclear. Overall, our findings suggest that young healthy adults may be more asymmetrical during fast treadmill walking than normal treadmill walking (see Table 3). In addition, our results revealed standard deviations of the left side higher than the right side for hip, knee, and ankle motions during normal and fast treadmill walking. A possible explanation for this difference may be related to laterality [41], as in our study, 52 out 60 participants reported to be right-dominant, with leg dominance being defined as the preferred leg for kicking a ball. Even though previous reports indicate that walking slowly is more challenging to the motor control of gait than usual and faster speed walks [42, 43], differences of gait motor control between usual and faster speed walking are not clear. Therefore, the findings of this study reported as the degree of asymmetry and the normalized symmetry index may be useful indicators of the gait motor control at different walking speeds.

Previous studies have investigated spine and lower body gait kinematics [6, 8, 9, 24–27]. In addition, although several studies have investigated on gait symmetry [17–23, 44] and presented valuable information, there is not generally accepted standard for characterization of gait symmetry [23]. Asymmetric gait patterns in healthy individuals may be expected as there exist natural functional differences between the lower extremities [12, 41, 42], such as the contribution of each limb in carrying out the tasks of propulsion and control during able-bodied

walking [41]. The present study provides information not only on the degree of asymmetry, the mean angular difference between left and right sides, but also on the $SI_{norm}$ in healthy individuals during normal and fast treadmill walking. Such information will add to the knowledge provided by previous investigations to better understand spine, pelvis, and lower body motions in healthy individuals. Our findings on $SI_{norm}$ for lower body motions in the sagittal plane were comparable to the ones described in [23]. Moreover, this study adds information on the $SI_{norm}$ parameter by describing the spine and pelvis 3D angular motions. In addition, reference degree of asymmetry information in healthy individuals has been presented in this study to help in the biomechanical assessment pathological individuals. Although the use of this indicator may be confusing as it is not referenced to the joint range of motion, such indicator has been implemented to assess asymmetry in pathological individuals. For instance, the degrees of asymmetry reported in total hip [16, 28, 29] and knee [15] replacement patients are greater than the degrees of asymmetry observed in the present study. Consequently, degrees of asymmetry greater than the ones reported in this study may be an indicative of abnormal gait function.

Several limitations need to be considered to interpret the present results. To begin with, the average age of the male and female participants in this study was ~21 years old, 52 out of 60 participants reported to be right-dominant, and all participants reported a healthy lifestyle (exercised at least twice a week); hence, results may be limited to similar populations. Furthermore, few gait cycles (~30) were used in normal and fast treadmill walking conditions; hence, the long terms of gait asymmetry kinematics were not explored. Additionally, participants wore different types of shoes during the experiments; thus, the influence of distinct shoes was not investigated in this study. Moreover, the skin-marker-based tracking technique used in this study is vulnerable to soft tissue artifacts [45]; however, clusters of at least four markers were used in each segment to reduce the influence of soft tissue artifacts. In addition, no ground reaction force, or electromyography (EMG) data was used in this study and thus, neither body kinetics, nor muscle activation patterns, were included. Future studies should include joint kinetics, ground reaction forces, and EMG data to gain a better understanding of asymmetry patterns in gait biomechanics.

## Conclusions

The present study revealed significant asymmetries in upper lumbar, lower lumbar, and pelvis segments, as well as in hip, knee, and ankle joints during normal and fast treadmill walking. Few asymmetry patterns were similar between normal and fast treadmill walking, whereas others appeared either only during normal or fast treadmill walking. Our findings suggest that young healthy adults may be more asymmetrical during fast treadmill walking than normal treadmill walking. The current study methodology allows for observation of asymmetries throughout the gait cycle and introduces reference values based on two symmetry indicators. These findings could provide insights into better understanding gait asymmetry in healthy individuals, and use them as reference indicators in diagnosing and evaluating abnormal gait function.

## Author Contributions

**Conceptualization:** Paul Gonzalo Arauz, Maria-Gabriela Garcia, Bernard Martin.

**Data curation:** Paul Gonzalo Arauz, Maria-Gabriela Garcia, Patricio Chiriboga, Sebastian Taco-Vasquez, Diego Klaic.

**Formal analysis:** Paul Gonzalo Arauz, Maria-Gabriela Garcia, Patricio Chiriboga, Sebastian Taco-Vasquez, Diego Klaic, Emilia Verdesoto.

**Project administration:** Paul Gonzalo Arauz.

**Visualization:** Paul Gonzalo Arauz, Bernard Martin.

**Writing – original draft:** Paul Gonzalo Arauz, Maria-Gabriela Garcia, Patricio Chiriboga, Sebastian Taco-Vasquez, Diego Klaic, Emilia Verdesoto, Bernard Martin.

**Writing – review & editing:** Paul Gonzalo Arauz, Maria-Gabriela Garcia, Patricio Chiriboga, Sebastian Taco-Vasquez, Bernard Martin.

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
