## [Decision Letter · Decision Letter 0]

12 May 2022

PONE-D-22-08871Spine and lower body motion symmetry during treadmill walking in healthy individuals - In-vivo 3-dimensional kinematic analysisPLOS ONE

Dear Dr. Arauz,

Thank you for submitting your manuscript to PLOS ONE. After careful consideration, we feel that it has merit but does not fully meet PLOS ONE’s publication criteria as it currently stands. Therefore, we invite you to submit a revised version of the manuscript that addresses the points raised during the review process.

You will see from the two reviews below that the manuscript requires more clarity, especially in defining relevant terms. Additionally, the 'real-life' application as well as the addition to the field of the present study in comparison to previous literature (or how the previous literature is improved with the present study) needs to be made explicit. I invite you to carefully consider the comments made and address them carefully. Finally, please make available the data set used or provide a justification for not doing so.  

We look forward to receiving your revised manuscript.

Kind regards,

Theodoros M. Bampouras

Academic Editor

PLOS ONE

Journal Requirements:

Reviewers' comments:

Reviewer's Responses to Questions

**Comments to the Author**

1. Is the manuscript technically sound, and do the data support the conclusions?

Reviewer #1: Partly

Reviewer #2: Partly

2. Has the statistical analysis been performed appropriately and rigorously? 

Reviewer #1: No

Reviewer #2: Yes

3. Have the authors made all data underlying the findings in their manuscript fully available?

Reviewer #1: No

Reviewer #2: No

4. Is the manuscript presented in an intelligible fashion and written in standard English?

Reviewer #1: No

Reviewer #2: Yes

5. Review Comments to the Author

Reviewer #1: Overall, a nice publication on gait symmetry in healthy individuals. Gait symmetry is an interesting parameter for diagnostics and therapy control. The presented methods seem suitable for further analysis. Nevertheless, a detailed explanation of the method and the comparison with other methods and results for gait symmetry is missing. The argumentation presented in the introduction and discussion included good arguments, underlined by references. But, there already exists some literature about gait symmetry in patients and healthy individuals, which has to be mentioned. The state of the art is missing and the purpose of this new study is not clear.

The gait symmetry, as the described result, is presented on over 10 pages. Discussed were differences between the degree of asymmetry, which is questionable, as this parameter is not referenced to the different range of motions of different movements. In addition, differences in gait symmetry between normal and fast walking were discussed, but not statistically analyzed. The most relevant question, which parameter defines the symmetry is not mentioned and discussed in comparison with existing literature. Moreover, comparing the gait symmetry during different movements, first, the gait symmetry in different participants should be analyzed. All in all, the goal of the paper is not clear.

In addition, some words are used confusingly. For example, talking about motion analysis ‘rotation’ is used to describe the movement of a joint in one plane, often in the transversal plane. In contrast, in the presented manuscript rotation is also used to talk about the general motion in all planes. Be careful to use the common vocabulary of the field. Other vocabularies are mentioned in the detailed comments.

In conclusion, the analysis of gait symmetry based on objective motion parameters is a rising and important field, which can be supported by the presented methods. Nevertheless, the manuscript needs to be mainly restructured and is not acceptable in its current form.

Detailed corrections and comments can be found below:

Abstract:

• Missing the description of how symmetry was calculated.

• Moreover, it is confusing, first talking only about rotation and mentioning afterward also tilt and knee flexion. See the comment in the summary for the usage of the word ‘rotation’.

• Line 37: I think it should be ‘rotation’ and not ‘notation’.

• Line 38: What is meant by upper ipsilateral tilt? What is the joint you are talking about?

Introduction:

• Gait symmetry was calculated in different ways, but there is no information about the way how to calculate gait symmetry.

• Last paragraph: Again the question arises, why you are talking only about joint rotations? 3D movements are described in most cases as flexion-extension; abduction-adduction and internal-external rotation. Therefore it is confusing to talk only about rotations if 3D kinematics will be analyzed.

Methods:

• Line 82: What are most participants? And how did you control the dominant leg?

• Procedure and data collection: What about a calibration measurement?

• Line 149-50: Is there a reference about the procedure you used for heel-contact and toe-off detection? In addition, I would be happy about some more information about this procedure of gait event detection.

• Statistical analysis:

o Only the SPM was described and no other statistical tests which were used.

o In the references, the SPM with the help of python was described in contrast to the mentioned software MATLAB. I would prefer more suitable references.

o Moreover, the threshold, mentioned in the results section, is not explained.

Results:

• Line 166: What is meant by ‘spine gait’? I would talk about asymmetric spine motion during normal treadmill walking.

• Line 170: What is meant by the degree of asymmetry averages? Is it possible to describe or visualize how this parameter is defined?

• Line 189: Please explain the threshold you describe.

• Line 189-90: Left side angles cannot be greater and lesser than right side angles at the same time, they can be greater or lesser.

• Line 199: Again, ‘spine gait’?

• Line 202: It should be ‘asymmetrical motion’ or ‘no asymmetry’.

• Line 218: Again threshold?

• Line 222 and 262: What is meant by ‘lower body gait’? Lower body or lower limb motion sounds better.

• It is not necessary to mention the results in the text and the table. A reference to the table is enough.

Discussion:

• Line 300: I think it should be ‘hip internal rotation’ and not ‘notation’.

• Line 301: Be precise, what ‘upper ipsilateral tilt’? Talking about the upper lumbar ipsilateral tilt is understandable.

• Comparing the symmetry or asymmetry of normal and fast treadmill walking is questionable if the gait symmetry between the participants is not analyzed.

• Line 318-19: There exist some studies analyzing gait symmetry of healthy participants, for example:

o Cimolin V, Cau N, Sartorio A, et al. Symmetry of Gait in Underweight, Normal and Overweight Children and Adolescents. Sensors (Basel) 2019; 19:9; doi:10.3390/s19092054.

o Nigg S, Vienneau J, Maurer C, Nigg BM. Development of a symmetry index using discrete variables. Gait Posture 2013; 38:1; doi:10.1016/j.gaitpost.2012.10.024.

o Queen R, Dickerson L, Ranganathan S, Schmitt D. A novel method for measuring asymmetry in kinematic and kinetic variables: The normalized symmetry index. J Biomech 2020; 99; doi:10.1016/j.jbiomech.2019.109531.

o Gouwanda D. Further validation of Normalized Symmetry Index and normalized cross-correlation in identifying gait asymmetry on restricted knee and ankle movement. IEEE-EMBS Conference on Biomedical Engineering and Sciences 2012; doi:10.1109/IECBES.2012.6498167.

o Herzog W, Nigg BM, Read LJ, Olsson E. Asymmetries in ground reaction force patterns in normal human gait. Med Sci Sports Exerc 1989; 21:1; doi:10.1249/00005768-198902000-00020.

o Gouwanda D, Senanayake SMNA. Identifying gait asymmetry using gyroscopes--a cross-correlation and Normalized Symmetry Index approach. J Biomech 2011; 44:5; doi:10.1016/j.jbiomech.2010.12.013.

o Gouwanda D, Senanayake SMNA. Periodical gait asymmetry assessment using real-time wireless gyroscopes gait monitoring system. J Med Eng Technol 2011; 35:8; doi:10.3109/03091902.2011.627080.

o Xu Y, Hou Q, Wang C, Simpson T, Bennett B, Russell S. How Well Can Modern Nonhabitual Barefoot Youth Adapt to Barefoot and Minimalist Barefoot Technology Shoe Walking, in regard to Gait Symmetry. Biomed Res Int 2017; 2017; doi:10.1155/2017/4316821.

o Siebers, HL, Alrawashdeh, W, Migliorini, F, Hildebrand, F, Betsch, M, Eschweiler, J, Comparison of different symmetry indices for the quantification of dynamic joint angles. BMC Sports Science, Medicine and Rehabilitation, 2021.

o Alrawashdeh, W, Siebers, HL, Reim, J, Rath, B, Tingart, M, Eschweiler, J, Gait symmetry – A valid parameter for pre and post planning for total knee arthroplasty. Journal of Musculoskeletal and Neuronal Interactions, 2021.

• Line 324-26: The argument was already presented in the introduction.

• In Figure 1 different participants were shown, with different shoes. The influence of different shoes was not mentioned in the limitation section.

Overall:

• The ‘degree of asymmetry’ is presented in degree and described as a mean difference between left and right side movements. Comparing this degree of asymmetry between different motions is difficult, as it should be interpreted in combination with the range of motion. 3° difference is a lot in case of a range of motion of 10° for posterior-anterior pelvic tilt. In contrast in the case of an 80° range of motion for knee flexion-extension, a 3° difference is negligible.

Figures:

• A grid in the graphs could be helpful.

• It is not necessary to mention a complex description two times, in the manuscript and the figure heading.

• Figures 5 & 6: For hip, knee, and ankle flexion-extension the standard deviation of the left side is much higher than from the right side, which should be mentioned and discussed.

Reviewer #2: Major comment:

While the manuscript is, in my opinion, generally well-written, and methodology does not pose any major problems, I think the discussion section needs to include much more detail on the interpretation of the results. There are a number of significant differences found between L/R sides for very subtle values of asymmetry, and I question what the real-life impact these values have or represent. Is a difference of <0.5 degrees noticeable? What does this mean for a healthy individual?

Specific comments:

Line 36 – ‘compared for treadmill walking’ – suggest adding in detail about normal and fast speeds

Line 67 – ‘common in clinical and rehabilitation practices’ – yes, sure, but these would be for injured or clinical populations, so why not just use overground walking for your healthy adults in this study? Or both overground and treadmill walking?

Line 73 – this hypothesis seems to be worded as if it is the null hypothesis?

Line 82 – what is meant by ‘right-leg dominant’? How was leg dominance defined? Is this relevant here?

Line 91-93 – I’m assuming the walking speeds were determined in separate trials (e.g. 4 x regular; 4 x fast) and not a within-trial acceleration?

Line 113 – ‘z axes’ – it would be helpful to state explicitly which direction this is (vertical)

Line 126-130 – were kinematics filtered at all?

Statistical analyses – (i) It appears that males and females were combined. Would there be reason to separate these two groups to evaluate for sex-based characteristics/differences? (ii) Why weren’t statistical analyses included to compare the treadmill walking speeds?

Line 171 – the values reported here (and throughout the rest of the results section) for detected degrees of asymmetry are extremely small (e.g. posterior tilt 0.46+/- 0.03 degree) and in many cases for a very short period of time (e.g. lower lumbar p/a tilt at normal speed – 2-3% gait cycle). Could they be prone to marker placement error? Or artefacts from the SPM analysis (e.g. as a result of temporal shifting – I don’t think this is the case, but more details on how SPM was performed in the methods section might also help)? With that in mind, are these results meaningful?

Line 300 – ‘internal notation’ – I think this is meant to be ‘rotation’?

Line 305 – links to comment above regarding hypothesis – this statement seems to conflict with the way the hypothesis was worded (original hypothesis suggests there would not be any differences, but you have found some, so null hypothesis (that there would be differences) is, at least, partially supported here).

Line 312 – again, why didn’t you make these comparisons (i.e. through statistical testing)?

Line 321 – some examples of the differences to which you refer here would be helpful

Line 333 – the wording of this limitation isn’t clear to me; what do you mean by ‘few gait cycles with repetitions’?

Line 345 – again, you didn’t run any tests so I think this statement needs to address that.

Line 347 – as per my main comment with the discussion section, a summary statement of the significance of these very small degrees of asymmetry should be included. What is the take-home message/impact of these findings?

Figure 3 – legend for Upper Lumbar P/A tilt has ‘GWN’ which isn’t explained anywhere. I’m assuming it is ‘ground walking normal’, but might be a typo?

Whilst the authors have indicated that the data are available, I do not see a link to where these are provided?

6. PLOS authors have the option to publish the peer review history of their article (what does this mean?). If published, this will include your full peer review and any attached files.

Reviewer #1: No

Reviewer #2: No

---

## [Author Response · Author response to Decision Letter 0]

22 May 2022

AUTHOR’S RESPONSE TO REVIEWER’S COMMENTS 5/22/2022

The authors wish to thank the Editor and Reviewers for their invaluable suggestions. The manuscript has been revised according to the editor’s and reviewers’ comments. A point-by-point reply is given below. The line, figure and table numbers noted in the replies refer to the revised manuscript. In the following response, the editor’s and reviewer’s comments are shown in italics and our responses appear immediately below in normal type.

Editor:

Journal Requirements:

Reply: We thank the Editor for the comments. We have formatted the main body and title of the manuscript accordingly to meet PLOS ONE’s style requirements.

Reply: We thank the Editor for the comment. Temporarily, we have made our data available at the following link

https://estudusfqedu-my.sharepoint.com/:f:/g/personal/parauz_usfq_edu_ec/EhXWpdGUuiBHsoomPt2-LYMBo54z04eM9wKcmxkeWGQVeA?e=aSeMbS

However, upon acceptance of the manuscript, we will make all our data available in a public repository.

Reply: We thank the Editor for the comment. Temporarily, we have made our data available at the following link

https://estudusfqedu-my.sharepoint.com/:f:/g/personal/parauz_usfq_edu_ec/EhXWpdGUuiBHsoomPt2-LYMBo54z04eM9wKcmxkeWGQVeA?e=aSeMbS

However, upon acceptance of the manuscript, we will make all our data available at a public repository.

In addition, we have explained the data availability information in our cover letter. 

Reviewer #1

Reviewer #1: Overall, a nice publication on gait symmetry in healthy individuals. Gait symmetry is an interesting parameter for diagnostics and therapy control. The presented methods seem suitable for further analysis. Nevertheless, a detailed explanation of the method and the comparison with other methods and results for gait symmetry is missing. The argumentation presented in the introduction and discussion included good arguments, underlined by references. But, there already exists some literature about gait symmetry in patients and healthy individuals, which has to be mentioned. The state of the art is missing and the purpose of this new study is not clear.

Reply: We thank and acknowledge the Reviewer for the comments. The Introduction and Discussion Sections have been revised accordingly in order to increase clarity for readership. A more detailed state of the art has been introduced and discussed, and the purpose of the study has been clarified.

Line 62-74:

Gait symmetry has been suggested as an important indicator of gait function in impaired and healthy individuals [14-16]. Consequently, better implementation and evaluation of restorative interventions requires appropriate assessment and characterization of normative gait symmetry in healthy individuals. Several studies have investigated on gait symmetry [17-23], as well as on spine and lower body kinematics during gait [6, 8, 9, 24-27]. Despite such contributions, up to date, there is no generally accepted standard for assessing symmetry [23]. Typically, the symmetry index, symmetry ratio, and statistical approaches are implemented to determine gait symmetry [21]. Thus, it is complicated to compare among studies and establish standard criteria to guide clinical decision-making. Several studies have implemented statistical procedures to investigate interlimb asymmetries using the mean difference between left and right limbs as a symmetry parameter in pathological individuals [15, 16, 28, 29]. However, implementation of such approaches to determine reference degree of asymmetry information in healthy individuals is lacking.

Line 80-88:

Therefore, the purpose of the present study was to analyze 3D spine, pelvis, and lower body gait symmetry kinematics during normal and fast treadmill walking in healthy individuals. This study applies statistical parametric mapping (SPM) [12], detecting significant differences between left and right-side movements, as well as the normalized symmetry index [21, 23] approaches to determine spine, pelvis, and lower body asymmetries. We hypothesized that there are significant differences between the spine and pelvis angular movements associated with the left lower limb motions and spine and pelvis angular movements associated with the right lower limb motions, as well as between the left and right lower extremity joint angles during normal and fast treadmill walking in healthy individuals.

Line 374-394:

Previous studies have investigated spine and lower body gait kinematics [6, 8, 9, 24-27]. In addition, although several studies have investigated on gait symmetry [17-23, 45] and presented valuable information, there is not generally accepted standard for characterization of gait symmetry [23]. Asymmetric gait patterns in healthy individuals may be expected as there exist natural functional differences between the lower extremities [12, 41, 43], such as the contribution of each limb in carrying out the tasks of propulsion and control during able-bodied walking [43]. The present study provides information not only on the degree of asymmetry, the mean angular difference between left and right sides, but also on the SInorm in healthy individuals during normal and fast treadmill walking. Such information will add to the knowledge provided by previous investigations to better understand spine, pelvis, and lower body motions in healthy individuals. Our findings on SInorm for lower body motions in the sagittal plane were comparable to the ones described in [23]. Moreover, this study adds information on the SInorm parameter by describing the spine and pelvis 3D angular motions. In addition, reference degree of asymmetry information in healthy individuals has been presented in this study to help in the biomechanical assessment pathological individuals. Although the use of this indicator may be confusing as it is not referenced to the joint range of motion, such indicator has been implemented to assess asymmetry in pathological individuals. For instance, the degrees of asymmetry reported in total hip [16, 28, 29] and knee [15] replacement patients are greater than the degrees of asymmetry observed in the present study. Consequently, degrees of asymmetry greater than the ones reported in this study may be an indicative of abnormal gait function.

The gait symmetry, as the described result, is presented on over 10 pages. Discussed were differences between the degree of asymmetry, which is questionable, as this parameter is not referenced to the different range of motions of different movements. In addition, differences in gait symmetry between normal and fast walking were discussed, but not statistically analyzed. The most relevant question, which parameter defines the symmetry is not mentioned and discussed in comparison with existing literature. Moreover, comparing the gait symmetry during different movements, first, the gait symmetry in different participants should be analyzed. All in all, the goal of the paper is not clear.

Reply: We agree and thank the Reviewer for the insightful comments. Even though the degree of asymmetry is not referenced to the different ranges of motions of different movements, it is a parameter that has been reported in the literature, for example:

1. Arauz P, Peng Y, Kwon Y-M. Knee motion symmetry was not restored in patients with unilateral bi-cruciate retaining total knee arthroplasty—in vivo three-dimensional kinematic analysis. International orthopaedics 42(12): 2817, 2018

2. Arauz P, Peng Y, MacAuliffe J, Kwon Y-M. In-vivo 3-Dimensional gait symmetry analysis in patients with bilateral total hip arthroplasty. Journal of biomechanics 77: 131, 2018

3. Tsai T-Y, Dimitriou D, Li J-S, Woo Nam K, Li G, Kwon Y-M. Asymmetric hip kinematics during gait in patients with unilateral total hip arthroplasty: In vivo 3-dimensional motion analysis. Journal of Biomechanics 48(4): 555, 2015

Nevertheless, in order to increase clarity for readership, in addition to the degree of asymmetry, the normalized symmetry index has been calculated and discussed accordingly. Figures 3, 4, 5, and 6, as well as Tables 1 and 2 have been revised accordingly. Likewise, Table 3, describing the result of the normalized symmetry index has been included.

Furthermore, the information from the normalized symmetry index has been used to statistically analyze the differences between normal and fast treadmill walking.

The degree of asymmetry and the normalized symmetry index have been described and discussed accordingly to make the paper clearer.

 The purpose of the study has been revised to increase the clarity for readership.

Line 170-174:

Symmetry was calculated throughout the gait cycle for spine, pelvis, and lower body motions. Statistical parametric mapping and the normalized symmetry index, presented by Gouwanda et al. [21], were calculated for assessing gait symmetry of the spine and pelvis angular motions, as well as lower body joint angles. The normalized symmetry index (SInorm) was calculated based on Eq. 1 [19-21, 23].

Line 333-342:

Descriptive statistics of the SInorm and its comparison between normal and fast treadmill walking is presented in Table 3. Overall, greater asymmetries were found during fast treadmill walking than normal treadmill walking.

Table 3. Descriptive statistics of the maximum SInorm in % and its comparison between normal and fast treadmill walking for N=60 participants.

Segment Motion Treadmill Walking Normal Treadmill Walking Fast 

 Mean SD Max Min Mean SD Max Min p-value

Upper Thorax Posterior-Anterior Tilt 45.56 12.47 66.67 12.70 44.17 12.61 65.61 12.64 0.201

 Ipsi-Contra Lateral Tilt 13.87 11.07 60.71 0.56 15.36 12.07 63.82 0.84 0.109

 Contra-Ipsi Lateral Rotation 22.28 14.08 64.70 0.81 22.59 12.30 58.93 1.28 0.765

Lower Thorax Posterior-Anterior Tilt 44.88 11.76 66.50 15.60 42.42 11.28 65.84 9.50 0.019

 Ipsi-Contra Lateral Tilt 15.11 11.30 56.47 0.46 17.82 12.17 59.03 0.00 0.000

 Contra-Ipsi Lateral Rotation 13.60 9.13 60.25 2.03 13.69 10.75 66.07 0.54 0.919

Upper Lumbar Posterior-Anterior Tilt 42.05 11.84 66.67 9.69 41.35 11.09 66.67 16.63 0.422

 Ipsi-Contra Lateral Tilt 13.77 10.86 64.38 0.00 13.65 10.10 59.13 0.18 0.867

 Contra-Ipsi Lateral Rotation 10.87 8.21 64.26 1.05 11.42 9.24 66.67 0.41 0.456

Lower Lumbar Posterior-Anterior Tilt 37.29 12.85 66.67 11.15 37.79 12.95 66.67 12.54 0.574

 Ipsi-Contra Lateral Tilt 14.82 10.16 55.32 0.00 15.90 11.83 59.77 0.59 0.163

 Contra-Ipsi Lateral Rotation 14.51 9.99 56.92 0.05 16.82 11.77 63.86 0.75 0.002

Pelvis Posterior-Anterior Tilt 38.10 14.02 66.67 6.40 37.77 13.83 66.67 4.94 0.770

 Ipsi-Contra Lateral Tilt 9.83 10.20 65.57 0.28 10.49 11.56 62.98 0.05 0.474

 Contra-Ipsi Lateral Rotation 12.25 8.51 50.01 0.67 11.88 8.95 59.82 0.29 0.602

Hip Flexion-Extension 7.98 10.13 65.29 0.15 8.18 8.86 60.56 0.33 0.787

 Adduction-Abduction 15.39 15.10 66.67 0.84 16.40 14.69 66.62 1.59 0.372

 Internal-External Rotation 28.46 15.05 66.06 3.25 29.51 13.59 63.38 5.99 0.164

Knee Flexion-Extension 6.62 6.64 61.39 0.63 7.71 6.19 62.62 0.40 0.045

 Adduction-Abduction 29.80 16.00 66.05 2.38 30.33 13.42 63.55 2.61 0.545

 Internal-External Rotation 30.06 13.17 62.85 6.59 27.71 12.44 62.08 4.34 0.009

Ankle Dorsi-Plantar Flexion 11.77 9.08 66.67 0.64 10.91 7.88 58.57 1.045 0.294

 Eversion-Inversion 21.22 10.39 57.16 1.87 22.97 11.76 63.11 1.98 0.036

 Internal-External Rotation 30.00 14.85 66.61 1.17 31.13 14.18 66.67 1.22 0.261

Abbreviations: SD, standard deviation; Max, maximum; Min, minimum.

Line 374-394:

Previous studies have investigated spine and lower body gait kinematics [6, 8, 9, 24-27]. In addition, although several studies have investigated on gait symmetry [17-23, 45] and presented valuable information, there is not generally accepted standard for characterization of gait symmetry [23]. Asymmetric gait patterns in healthy individuals may be expected as there exist natural functional differences between the lower extremities [12, 41, 43], such as the contribution of each limb in carrying out the tasks of propulsion and control during able-bodied walking [43]. The present study provides information not only on the degree of asymmetry, the mean angular difference between left and right sides, but also on the SInorm in healthy individuals during normal and fast treadmill walking. Such information will add to the knowledge provided by previous investigations to better understand spine, pelvis, and lower body motions in healthy individuals. Our findings on SInorm for lower body motions in the sagittal plane were comparable to the ones described in [23]. Moreover, this study adds information on the SInorm parameter by describing the spine and pelvis 3D angular motions. In addition, reference degree of asymmetry information in healthy individuals has been presented in this study to help in the biomechanical assessment pathological individuals. Although the use of this indicator may be confusing as it is not referenced to the joint range of motion, such indicator has been implemented to assess asymmetry in pathological individuals. For instance, the degrees of asymmetry reported in total hip [16, 28, 29] and knee [15] replacement patients are greater than the degrees of asymmetry observed in the present study. Consequently, degrees of asymmetry greater than the ones reported in this study may be an indicative of abnormal gait function.

In addition, some words are used confusingly. For example, talking about motion analysis ‘rotation’ is used to describe the movement of a joint in one plane, often in the transversal plane. In contrast, in the presented manuscript rotation is also used to talk about the general motion in all planes. Be careful to use the common vocabulary of the field. Other vocabularies are mentioned in the detailed comments.

Reply: We agree and thank the Reviewer for the comments. All words and terminology have been revised accordingly to make the manuscript clearer.

In conclusion, the analysis of gait symmetry based on objective motion parameters is a rising and important field, which can be supported by the presented methods. Nevertheless, the manuscript needs to be mainly restructured and is not acceptable in its current form.

Reply: We agree and thank the Reviewer for the comments. All comments have been addressed, and the manuscript has been revised and restructured accordingly in order to increase the clarity for readership. 

Detailed corrections and comments can be found below:

Abstract:

• Missing the description of how symmetry was calculated.

Reply: We agree and thank the Reviewer for the insightful comment. The description of the approaches used to determine symmetry have been described accordingly in Abstract.

Line 34-36:

Statistical parametric mapping and the normalized symmetry index approaches were used to determine spine, pelvis, and lower body asymmetries during treadmill walking.

• Moreover, it is confusing, first talking only about rotation and mentioning afterward also tilt and knee flexion. See the comment in the summary for the usage of the word ‘rotation’.

Reply: We agree and thank the Reviewer for the comments. All words and terminology have been revised accordingly to make the manuscript clearer.

Line 36-41:

The spine and pelvis angular motions associated with the left and right lower limb motions, as well as the left and right lower extremity joint angles were compared for normal and fast treadmill walking. The lower lumbar contra-ipsi lateral rotation (5.74±0.04°) and hip internal rotation (5.33±0.18°) presented the largest degrees of asymmetry during normal treadmill. Upper lumbar ipsi-contra lateral tilt (1.48±0.14°) and knee flexion (2.98±0.13°) indicated the largest asymmetries and during fast treadmill walking.

• Line 37: I think it should be ‘rotation’ and not ‘notation’.

Reply: We agree and thank the Reviewer for the comments. All words and terminology have been revised accordingly to make the manuscript clearer.

Line 38-41:

The lower lumbar contra-ipsi lateral rotation (5.74±0.04°) and hip internal rotation (5.33±0.18°) presented the largest degrees of asymmetry during normal treadmill. Upper lumbar ipsi-contra lateral tilt (1.48±0.14°) and knee flexion (2.98±0.13°) indicated the largest asymmetries and during fast treadmill walking.

• Line 38: What is meant by upper ipsilateral tilt? What is the joint you are talking about?

Reply: We thank the Reviewer for the comment. The sentence has been revised accordingly to make the manuscript clearer.

Line 38-41:

The lower lumbar contra-ipsi lateral rotation (5.74±0.04°) and hip internal rotation (5.33±0.18°) presented the largest degrees of asymmetry during normal treadmill. Upper lumbar ipsi-contra lateral tilt (1.48±0.14°) and knee flexion (2.98±0.13°) indicated the largest asymmetries and during fast treadmill walking.

Introduction:

• Gait symmetry was calculated in different ways, but there is no information about the way how to calculate gait symmetry.

Reply: We agree and thank the Reviewer for the comment. Information on the approaches used to calculate gait symmetry has been included in the Introduction Section accordingly in order to increase clarity for readership.

Line 65-74:

Several studies have investigated on gait symmetry [17-23], as well as on spine and lower body kinematics during gait [6, 8, 9, 24-27]. Despite such contributions, up to date, there is no generally accepted standard for assessing symmetry [23]. Typically, the symmetry index, symmetry ratio, and statistical approaches are implemented to determine gait symmetry [21]. Thus, it is complicated to compare among studies and establish standard criteria to guide clinical decision-making. Several studies have implemented statistical procedures to investigate interlimb asymmetries using the mean difference between left and right limbs as a symmetry parameter in pathological individuals [15, 16, 28, 29]. However, implementation of such approaches to determine reference degree of asymmetry information in healthy individuals is lacking.

Line 80-84:

Therefore, the purpose of the present study was to analyze 3D spine, pelvis, and lower body gait symmetry kinematics during normal and fast treadmill walking in healthy individuals. This study applies statistical parametric mapping (SPM) [12], detecting significant differences between left and right-side movements, as well as the normalized symmetry index [21, 23] approaches to determine spine, pelvis, and lower body asymmetries.

• Last paragraph: Again the question arises, why you are talking only about joint rotations? 3D movements are described in most cases as flexion-extension; abduction-adduction and internal-external rotation. Therefore it is confusing to talk only about rotations if 3D kinematics will be analyzed.

Reply: We agree and thank the Reviewer for the comments. All words, sentences, and terminology have been revised accordingly to make the manuscript clearer.

Line 84-88:

We hypothesized that there are significant differences between the spine and pelvis angular movements associated with the left lower limb motions and spine and pelvis angular movements associated with the right lower limb motions, as well as between the left and right lower extremity joint angles during normal and fast treadmill walking in healthy individuals.

Methods:

• Line 82: What are most participants? And how did you control the dominant leg?

Reply: We thank the Reviewer for the comment. We have made clear in the Methods Section that fifty-two participants reported to be right-dominant. This has been included as a study limitation as no action was taken to control the dominant leg. 

Line 94-95:

Fifty-two out of the sixty participants reported to be right-leg dominant (with leg dominance being defined as the preferred leg for kicking a ball).

Line 394-397:

Several limitations need to be considered to interpret the present results. To begin with, the average age of the male and female participants in this study was ~21 years old, 52 out of 60 participants reported to be right-dominant, and all participants reported a healthy lifestyle (exercised at least twice a week); hence, results may be limited to similar populations.

• Procedure and data collection: What about a calibration measurement?

Reply: We thank the Reviewer for the comment. We have made clear in the Methods Section that a standard Vicon calibration procedure was applied to determine the three-dimensional coordinated of each reflective spherical marker.

Line 111-112:

The standard Vicon calibration procedures were applied to determine the 3D coordinates of the reflective spherical markers.

• Line 149-50: Is there a reference about the procedure you used for heel-contact and toe-off detection? In addition, I would be happy about some more information about this procedure of gait event detection.

Reply: We agree thank the Reviewer for the comment. This sentence has been revised and referenced in the Methods Section accordingly in order to increase the clarity for readership.

Line 164-165:

Strides were defined to start with the initial contact and end with the following initial contact of one foot [23].

• Statistical analysis:

o Only the SPM was described and no other statistical tests which were used.

Reply: We agree and thank the Reviewer for the comment. All statistical test used in this study have been described accordingly in the Methods Section in order to increase the clarity for readership.

Line 179-186:

The software MATLAB (MathWorks, Inc., Natick, MA) was used to performed SPM [12, 36-38] analyses using scalar fields to determine significant differences between the spine and pelvis angular motions associated with the left lower limb motions, and spine and pelvis angular motions associated with the right lower limb motions, as well as between the left and right hip, knee, and ankles joint angles throughout the gait cycle. A Student’s t-test was used to compare maximum SInorm differences between normal and fast treadmill walking. Likewise, A Student’s t-test compared walking speeds for each condition. A significance level of α = 0.05 was used for the analysis.

o In the references, the SPM with the help of python was described in contrast to the mentioned software MATLAB. I would prefer more suitable references.

Reply: We thank the Reviewer for the comment. SPM was first used in python and then implemented in MATLAB. The references used are suitable because SPM methodology and software application is explained. One more reference was included to indicate SPM application with both python and MATLAB in order make the manuscript clearer for the reader.

 1. Pataky TC, Vanrenterghem J, Robinson MA. The probability of false positives in zero-dimensional analyses of one-dimensional kinematic, force and EMG trajectories. Journal of Biomechanics 49(9): 1468, 2016

Line 179-186:

The software MATLAB (MathWorks, Inc., Natick, MA) was used to performed SPM [12, 36-38] analyses using scalar fields to determine significant differences between the spine and pelvis angular motions associated with the left lower limb motions, and spine and pelvis angular motions associated with the right lower limb motions, as well as between the left and right hip, knee, and ankles joint angles throughout the gait cycle. A Student’s t-test was used to compare maximum SInorm differences between normal and fast treadmill walking. Likewise, A Student’s t-test compared walking speeds for each condition. A significance level of α = 0.05 was used for the analysis.

o Moreover, the threshold, mentioned in the results section, is not explained.

Reply: We agree and thank the Reviewer for the insightful comment. The Results Section has been revised and the SPM threshold has been included accordingly. The SPM threshold is calculated automatically by using the SPM 1D function in MATALB. The threshold value must be exceeded to detect significant differences.

Line 193-194:

SPM analysis indicated that the upper and lower thorax segments presented symmetrical angular motions during normal treadmill walking, as the scalar field SPM curve did not exceed the threshold t* for α = 0.05 (Fig. 3).

Figure 3:

Results:

• Line 166: What is meant by ‘spine gait’? I would talk about asymmetric spine motion during normal treadmill walking.

Reply: We agree and thank the Reviewer for the comment. The sentence has been revised in the Results Section accordingly in order to increase clarity for readership.

Line 192:

Asymmetric spine motion during normal treadmill walking

• Line 170: What is meant by the degree of asymmetry averages? Is it possible to describe or visualize how this parameter is defined?

Reply: We agree and thank the Reviewer for the insightful comment. We have made clear in the Results Section that the average degree of asymmetry describes the mean difference between left and right-side movements when SPM detected significant differences.

Line 242-245:

Descriptive statistics of the average degree of asymmetry, describing the mean difference between left and right-side movements when the scalar field SPM detected significant differences, and the maximum magnitude of the SInorm when the scalar field SPM detected significant differences, in spine segments during normal and fast treadmill walking are presented in Table 1. 

• Line 189: Please explain the threshold you describe.

Reply: We agree and thank the Reviewer for the insightful comment. The Results Section has been revised and the SPM threshold has been included accordingly. The SPM threshold is calculated automatically by using the SPM 1D function in MATALB. The threshold value must be exceeded to detect significant differences.

Line 193-194:

SPM analysis indicated that the upper and lower thorax segments presented symmetrical angular motions during normal treadmill walking, as the scalar field SPM curve did not exceed the threshold t* for α = 0.05 (Fig. 3).

Figure 3:

• Line 189-90: Left side angles cannot be greater and lesser than right side angles at the same time, they can be greater or lesser.

Reply: We agree and thank the Reviewer for the comment. The sentence has been revised in the Results Section accordingly in order to increase clarity for readership.

Line 209-216:

Fig 3. Average and standard deviation of upper thorax, lower thorax, upper lumbar, and lower lumbar posterior-anterior (P/A) tilt, ipsi-contralateral (Ipsi/Con) tilt, and contra-ipsi (Con/Ipsi) rotation, for left and right sides during one gait cycle of normal treadmill walking (TWN) in sixty healthy participants. Green bars on the horizontal axis and the scalar field SPM results with threshold t* depict where, in % cycle, left side angles were greater or lesser than right side angles. The normalized symmetry index (SInorm) calculated during one gait cycle of TWN. Solid and dashed lines correspond to average left and right sides, as well as average SInorm, and shaded areas correspond to standard deviation. Black dotted vertical lines denote toe-off.

Figure 3:

• Line 199: Again, ‘spine gait’?

Reply: We agree and thank the Reviewer for the comment. The sentence has been revised in the Results Section accordingly in order to increase clarity for readership.

Line 218:

Asymmetric spine motions during fast treadmill walking

• Line 202: It should be ‘asymmetrical motion’ or ‘no asymmetry’.

Reply: We agree and thank the Reviewer for the comment. The sentence has been revised in the Results Section accordingly in order to increase clarity for readership.

Line 218-223:

SPM analysis showed that the upper and lower thorax segments presented symmetrical angular motions during fast treadmill walking (Fig. 4). SInorm values for upper and lower thorax posterior anterior tilt varied between ±36% whereas the ipsi-contra lateral tilt and contra-ipsi lateral rotation varied between ±17% (Fig. 4). SPM indicated that upper and lower lumbar angular motions were asymmetrical. The upper lumbar indicated a symmetrical posterior-anterior tilt, with SInorm values varying between ±25% (Fig. 4).

Figure 4:

• Line 218: Again threshold?

Reply: We agree and thank the Reviewer for the insightful comment. The Results Section has been revised and the SPM threshold has been included accordingly. The SPM threshold is calculated automatically by using the SPM 1D function in MATALB. The threshold value must be exceeded to detect significant differences. Fig. 4 illustrates examples of thresholds t* for different motions.

Line 124-128:

Average and standard deviation of upper thorax, lower thorax, upper lumbar, and lower lumbar posterior-anterior (P/A) tilt, ipsi-contralateral (Ipsi/Con) tilt, and contra-ipsi (Con/Ipsi) rotation, for left and right sides during one gait cycle of fast treadmill walking (TWF) in sixty healthy participants. Green bars on the horizontal axis and the scalar field SPM results with threshold t* depict where, in % cycle, left side angles were greater or lesser than right side angles. The normalized symmetry index (SInorm) calculated during one gait cycle of TWF. Solid and dashed lines correspond to average left and right sides, as well as average SInorm, and shaded areas correspond to standard deviation. Black dotted vertical lines denote toe-off.

Figure 4:

• Line 222 and 262: What is meant by ‘lower body gait’? Lower body or lower limb motion sounds better.

Reply: We agree and thank the Reviewer for the comment. The sentence has been revised in the Results Section accordingly in order to increase clarity for readership.

Line 257:

Asymmetric lower body motions during normal treadmill walking

• It is not necessary to mention the results in the text and the table. A reference to the table is enough.

Reply: We agree and thank the Reviewer for the comment. The Results Section has been revised accordingly to increase the clarity for the readership.

Line 242-245:

Descriptive statistics of the average degree of asymmetry, describing the mean difference between left and right-side movements when the scalar field SPM detected significant differences, and the maximum magnitude of the SInorm when the scalar field SPM detected significant differences, in spine segments during normal and fast treadmill walking are presented in Table 1.

Line 321-325:

Descriptive statistics of the average degree of asymmetry, describing the mean difference between left and right-side movements when the scalar field SPM detected significant differences, and the maximum magnitude of the SInorm when the scalar field SPM detected significant differences, in the pelvis segment and lower body joints during normal and fast treadmill walking are presented in Table 2.

Line 333-335:

Descriptive statistics of the SInorm and its comparison between normal and fast treadmill walking is presented in Table 3. Overall, greater asymmetries were found during fast treadmill walking than normal treadmill walking.

Discussion:

• Line 300: I think it should be ‘hip internal rotation’ and not ‘notation’.

Reply: We agree and thank the Reviewer for the comment. The sentence has been revised in the Discussion Section accordingly in order to increase clarity for readership.

Line 349-352:

Degrees of asymmetry and the associated maximum magnitude of SInorm of 5.74±0.04° and 14%, as well as 5.33±0.18° and 21%, for the lower lumbar contra-ipsi lateral rotation and hip internal rotation, respectively, were the largest asymmetries detected during normal treadmill walking.

• Line 301: Be precise, what ‘upper ipsilateral tilt’? Talking about the upper lumbar ipsilateral tilt is understandable.

Reply: We agree and thank the Reviewer for the comment. The sentence has been revised in the Discussion Section accordingly in order to increase clarity for readership.

Line 352-355:

Upper lumbar ipsi-contra lateral tilt and knee flexion-extension with degrees of asymmetry and the associated the maximum magnitude of SInorm of 1.48±0.14° and 15.3%, as well as 2.98±0.13° and 6.5%, respectively, were the largest asymmetries found during fast treadmill walking.

• Comparing the symmetry or asymmetry of normal and fast treadmill walking is questionable if the gait symmetry between the participants is not analyzed. 

Reply: We agree and thank the Reviewer for the insightful comment. The information from the normalized symmetry index has been used to statistically analyze the differences between normal and fast treadmill walking. Those results have been introduced and discussed accordingly. 

Line 333-342:

Descriptive statistics of the SInorm and its comparison between normal and fast treadmill walking is presented in Table 3. Overall, greater asymmetries were found during fast treadmill walking than normal treadmill walking.

Table 3. Descriptive statistics of the maximum SInorm in % and its comparison between normal and fast treadmill walking for N=60 participants.

Segment Motion Treadmill Walking Normal Treadmill Walking Fast 

 Mean SD Max Min Mean SD Max Min p-value

Upper Thorax Posterior-Anterior Tilt 45.56 12.47 66.67 12.70 44.17 12.61 65.61 12.64 0.201

 Ipsi-Contra Lateral Tilt 13.87 11.07 60.71 0.56 15.36 12.07 63.82 0.84 0.109

 Contra-Ipsi Lateral Rotation 22.28 14.08 64.70 0.81 22.59 12.30 58.93 1.28 0.765

Lower Thorax Posterior-Anterior Tilt 44.88 11.76 66.50 15.60 42.42 11.28 65.84 9.50 0.019

 Ipsi-Contra Lateral Tilt 15.11 11.30 56.47 0.46 17.82 12.17 59.03 0.00 0.000

 Contra-Ipsi Lateral Rotation 13.60 9.13 60.25 2.03 13.69 10.75 66.07 0.54 0.919

Upper Lumbar Posterior-Anterior Tilt 42.05 11.84 66.67 9.69 41.35 11.09 66.67 16.63 0.422

 Ipsi-Contra Lateral Tilt 13.77 10.86 64.38 0.00 13.65 10.10 59.13 0.18 0.867

 Contra-Ipsi Lateral Rotation 10.87 8.21 64.26 1.05 11.42 9.24 66.67 0.41 0.456

Lower Lumbar Posterior-Anterior Tilt 37.29 12.85 66.67 11.15 37.79 12.95 66.67 12.54 0.574

 Ipsi-Contra Lateral Tilt 14.82 10.16 55.32 0.00 15.90 11.83 59.77 0.59 0.163

 Contra-Ipsi Lateral Rotation 14.51 9.99 56.92 0.05 16.82 11.77 63.86 0.75 0.002

Pelvis Posterior-Anterior Tilt 38.10 14.02 66.67 6.40 37.77 13.83 66.67 4.94 0.770

 Ipsi-Contra Lateral Tilt 9.83 10.20 65.57 0.28 10.49 11.56 62.98 0.05 0.474

 Contra-Ipsi Lateral Rotation 12.25 8.51 50.01 0.67 11.88 8.95 59.82 0.29 0.602

Hip Flexion-Extension 7.98 10.13 65.29 0.15 8.18 8.86 60.56 0.33 0.787

 Adduction-Abduction 15.39 15.10 66.67 0.84 16.40 14.69 66.62 1.59 0.372

 Internal-External Rotation 28.46 15.05 66.06 3.25 29.51 13.59 63.38 5.99 0.164

Knee Flexion-Extension 6.62 6.64 61.39 0.63 7.71 6.19 62.62 0.40 0.045

 Adduction-Abduction 29.80 16.00 66.05 2.38 30.33 13.42 63.55 2.61 0.545

 Internal-External Rotation 30.06 13.17 62.85 6.59 27.71 12.44 62.08 4.34 0.009

Ankle Dorsi-Plantar Flexion 11.77 9.08 66.67 0.64 10.91 7.88 58.57 1.045 0.294

 Eversion-Inversion 21.22 10.39 57.16 1.87 22.97 11.76 63.11 1.98 0.036

 Internal-External Rotation 30.00 14.85 66.61 1.17 31.13 14.18 66.67 1.22 0.261

Abbreviations: SD, standard deviation; Max, maximum; Min, minimum.

Line 360-366:

It has been reported that the walking speed affects individuals’ gait kinematics [39, 40]; however, the influence of treadmill walking speed on gait symmetry kinematics in healthy individuals remains unclear. Even though it has been reported that the imposed constant speed of a treadmill may artificially impose motor control of gait and impede the natural variation that occurs during overground walking and therefore minimize gait variability [41, 42], overall, our findings suggest that young healthy adults may be more asymmetrical during fast treadmill walking than normal treadmill walking (see Table 3).

• Line 318-19: There exist some studies analyzing gait symmetry of healthy participants, for example:

o Cimolin V, Cau N, Sartorio A, et al. Symmetry of Gait in Underweight, Normal and Overweight Children and Adolescents. Sensors (Basel) 2019; 19:9; doi:10.3390/s19092054.

o Nigg S, Vienneau J, Maurer C, Nigg BM. Development of a symmetry index using discrete variables. Gait Posture 2013; 38:1; doi:10.1016/j.gaitpost.2012.10.024.

o Queen R, Dickerson L, Ranganathan S, Schmitt D. A novel method for measuring asymmetry in kinematic and kinetic variables: The normalized symmetry index. J Biomech 2020; 99; doi:10.1016/j.jbiomech.2019.109531.

o Gouwanda D. Further validation of Normalized Symmetry Index and normalized cross-correlation in identifying gait asymmetry on restricted knee and ankle movement. IEEE-EMBS Conference on Biomedical Engineering and Sciences 2012; doi:10.1109/IECBES.2012.6498167.

o Herzog W, Nigg BM, Read LJ, Olsson E. Asymmetries in ground reaction force patterns in normal human gait. Med Sci Sports Exerc 1989; 21:1; doi:10.1249/00005768-198902000-00020.

o Gouwanda D, Senanayake SMNA. Identifying gait asymmetry using gyroscopes--a cross-correlation and Normalized Symmetry Index approach. J Biomech 2011; 44:5; doi:10.1016/j.jbiomech.2010.12.013.

o Gouwanda D, Senanayake SMNA. Periodical gait asymmetry assessment using real-time wireless gyroscopes gait monitoring system. J Med Eng Technol 2011; 35:8; doi:10.3109/03091902.2011.627080.

o Xu Y, Hou Q, Wang C, Simpson T, Bennett B, Russell S. How Well Can Modern Nonhabitual Barefoot Youth Adapt to Barefoot and Minimalist Barefoot Technology Shoe Walking, in regard to Gait Symmetry. Biomed Res Int 2017; 2017; doi:10.1155/2017/4316821.

o Siebers, HL, Alrawashdeh, W, Migliorini, F, Hildebrand, F, Betsch, M, Eschweiler, J, Comparison of different symmetry indices for the quantification of dynamic joint angles. BMC Sports Science, Medicine and Rehabilitation, 2021.

o Alrawashdeh, W, Siebers, HL, Reim, J, Rath, B, Tingart, M, Eschweiler, J, Gait symmetry – A valid parameter for pre and post planning for total knee arthroplasty. Journal of Musculoskeletal and Neuronal Interactions, 2021.

Reply: We and thank the Reviewer for the insightful comment. The suggested studies have been referenced and discussed accordingly in order to increase the clarity for readership.

Line 374-393:

Previous studies have investigated spine and lower body gait kinematics [6, 8, 9, 24-27]. In addition, although several studies have investigated on gait symmetry [17-23, 45] and presented valuable information, there is not generally accepted standard for characterization of gait symmetry [23]. Asymmetric gait patterns in healthy individuals may be expected as there exist natural functional differences between the lower extremities [12, 41, 43], such as the contribution of each limb in carrying out the tasks of propulsion and control during able-bodied walking [43]. The present study provides information not only on the degree of asymmetry, the mean angular difference between left and right sides, but also on the SInorm in healthy individuals during normal and fast treadmill walking. Such information will add to the knowledge provided by previous investigations to better understand spine, pelvis, and lower body motions in healthy individuals. Our findings on SInorm for lower body motions in the sagittal plane were comparable to the ones described in [23]. Moreover, this study adds information on the SInorm parameter by describing the spine and pelvis 3D angular motions. In addition, reference degree of asymmetry information in healthy individuals has been presented in this study to help in the biomechanical assessment pathological individuals. Although the use of this indicator may be confusing as it is not referenced to the joint range of motion, such indicator has been implemented to assess asymmetry in pathological individuals. For instance, the degrees of asymmetry reported in total hip [16, 28, 29] and knee [15] replacement patients are greater than the degrees of asymmetry observed in the present study. Consequently, degrees of asymmetry greater than the ones reported in this study may be an indicative of abnormal gait function.

• Line 324-26: The argument was already presented in the introduction.

Reply: We agree and thank the Reviewer for the comment. The argument has been removed from the Discussion Section.

• In Figure 1 different participants were shown, with different shoes. The influence of different shoes was not mentioned in the limitation section.

Reply: We agree and thank the Reviewer for the comment. We have included in the Limitations Section that the influence of different shoes on gait symmetry was not investigated.

Line 399-401:

Additionally, participants wore different types of shoes during the experiments; thus, the influence of distinct shoes was not investigated in this study.

Overall:

• The ‘degree of asymmetry’ is presented in degree and described as a mean difference between left and right side movements. Comparing this degree of asymmetry between different motions is difficult, as it should be interpreted in combination with the range of motion. 3° difference is a lot in case of a range of motion of 10° for posterior-anterior pelvic tilt. In contrast in the case of an 80° range of motion for knee flexion-extension, a 3° difference is negligible.

Reply: We agree and thank the Reviewer for the insightful comment. Even though the degree of asymmetry is not referenced to the different range of motions of different movements, it is a parameter that has been reported in the literature, for example:

1. Arauz P, Peng Y, Kwon Y-M. Knee motion symmetry was not restored in patients with unilateral bi-cruciate retaining total knee arthroplasty—in vivo three-dimensional kinematic analysis. International orthopaedics 42(12): 2817, 2018

2. Arauz P, Peng Y, MacAuliffe J, Kwon Y-M. In-vivo 3-Dimensional gait symmetry analysis in patients with bilateral total hip arthroplasty. Journal of biomechanics 77: 131, 2018

3. Tsai T-Y, Dimitriou D, Li J-S, Woo Nam K, Li G, Kwon Y-M. Asymmetric hip kinematics during gait in patients with unilateral total hip arthroplasty: In vivo 3-dimensional motion analysis. Journal of Biomechanics 48(4): 555, 2015

Nevertheless, in order to increase clarity for readership, in addition to the degree of asymmetry, the normalized symmetry index has been calculated and discussed accordingly. Figures 3, 4, 5, and 6, as well as Tables 1 and 2 have been revised accordingly. Likewise, Table 3, describing the result of the normalized symmetry index has been included.

Line 242-245:

Descriptive statistics of the average degree of asymmetry, describing the mean difference between left and right-side movements when the scalar field SPM detected significant differences, and the maximum magnitude of the SInorm when the scalar field SPM detected significant differences, in spine segments during normal and fast treadmill walking are presented in Table 1.

Line 321-325:

Descriptive statistics of the average degree of asymmetry, describing the mean difference between Descriptive statistics of the average degree of asymmetry, describing the mean difference between left and right-side movements when the scalar field SPM detected significant differences, and the maximum magnitude of the SInorm when the scalar field SPM detected significant differences, in the pelvis segment and lower body joints during normal and fast treadmill walking are presented in Table 2.

Line 333-335:

Descriptive statistics of the SInorm and its comparison between normal and fast treadmill walking is presented in Table 3. Overall, greater asymmetries were found during fast treadmill walking than normal treadmill walking.

Figures:

• A grid in the graphs could be helpful.

Reply: We agree and thank the Reviewer for the comment. Several figures were revised. At the end, a grid was not included because figure information was clear enough.

Figure 4:

• It is not necessary to mention a complex description two times, in the manuscript and the figure heading.

Reply: We thank the Reviewer for the comment. Figure descriptions have been revised in the Results Section accordingly in order to increase the clarity for readership.

• Figures 5 & 6: For hip, knee, and ankle flexion-extension the standard deviation of the left side is much higher than from the right side, which should be mentioned and discussed.

Reply: We agree and thank the Reviewer for the insightful comment. We have made clear that the standard deviation of the left side is higher than from the right side in the Results and Discussion Sections accordingly in order to increase clarity for readership.

Line 275-276:

The standard deviation of the left side was higher than the right side for hip, knee, and ankle motions (Fig. 5).

Line 306-307:

The standard deviation of the left side was higher than the right side for hip, knee, and ankle motions (Fig. 6).

Line 366-372:

In addition, our results revealed standard deviations of the left side higher than the right side for hip, knee, and ankle motions during normal and treadmill walking. A possible explanation for this difference may be related to laterality [43], as in our study, 52 out 60 participants reported to be right-dominant, with leg dominance being defined as the preferred leg for kicking a ball. These results disagree with previous reports indicating that walking slowly is more challenging to the motor control of gait than usual and faster speed walks [41, 44].

Reviewer #2

Reviewer #2: Major comment:

While the manuscript is, in my opinion, generally well-written, and methodology does not pose any major problems, I think the discussion section needs to include much more detail on the interpretation of the results. There are a number of significant differences found between L/R sides for very subtle values of asymmetry, and I question what the real-life impact these values have or represent. Is a difference of <0.5 degrees noticeable? What does this mean for a healthy individual?

Reply: We agree and thank the Reviewer for the insightful comments. We have revised the Discussion Section in order to provide more details on the interpretation of the results. 

In addition to the degree of asymmetry, the normalized symmetry index has been calculated and discussed accordingly in order to increase the clarity for readership. Figures 3, 4, 5, and 6, as well as Tables 1 and 2 have been revised accordingly. Likewise, Table 3, describing the result of the normalized symmetry index has been included.

Even though the degree of asymmetry is not referenced to the different range of motions of different movements, it is a parameter that has been reported in the literature, for example:

1. Arauz P, Peng Y, Kwon Y-M. Knee motion symmetry was not restored in patients with unilateral bi-cruciate retaining total knee arthroplasty—in vivo three-dimensional kinematic analysis. International orthopaedics 42(12): 2817, 2018

2. Arauz P, Peng Y, MacAuliffe J, Kwon Y-M. In-vivo 3-Dimensional gait symmetry analysis in patients with bilateral total hip arthroplasty. Journal of biomechanics 77: 131, 2018

3. Tsai T-Y, Dimitriou D, Li J-S, Woo Nam K, Li G, Kwon Y-M. Asymmetric hip kinematics during gait in patients with unilateral total hip arthroplasty: In vivo 3-dimensional motion analysis. Journal of Biomechanics 48(4): 555, 2015

Furthermore, the information from the normalized symmetry index has been used to statistically analyze the differences between normal and fast treadmill walking.

The degree of asymmetry and the normalized symmetry index have been described and discussed accordingly to make the paper clearer.

The symmetry parameters presented in this study can be used as reference information to help to guide clinical decision-making. 

Line 170-174:

Symmetry was calculated throughout the gait cycle for spine, pelvis, and lower body motions. Statistical parametric mapping and the normalized symmetry index, presented by Gouwanda et al. [21], were calculated for assessing gait symmetry of the spine and pelvis angular motions, as well as lower body joint angles. The normalized symmetry index (SInorm) was calculated based on Eq. 1 [19-21, 23].

Line 333-342:

Descriptive statistics of the SInorm and its comparison between normal and fast treadmill walking is presented in Table 3. Overall, greater asymmetries were found during fast treadmill walking than normal treadmill walking.

Table 3. Descriptive statistics of the maximum SInorm in % and its comparison between normal and fast treadmill walking for N=60 participants.

Segment Motion Treadmill Walking Normal Treadmill Walking Fast 

 Mean SD Max Min Mean SD Max Min p-value

Upper Thorax Posterior-Anterior Tilt 45.56 12.47 66.67 12.70 44.17 12.61 65.61 12.64 0.201

 Ipsi-Contra Lateral Tilt 13.87 11.07 60.71 0.56 15.36 12.07 63.82 0.84 0.109

 Contra-Ipsi Lateral Rotation 22.28 14.08 64.70 0.81 22.59 12.30 58.93 1.28 0.765

Lower Thorax Posterior-Anterior Tilt 44.88 11.76 66.50 15.60 42.42 11.28 65.84 9.50 0.019

 Ipsi-Contra Lateral Tilt 15.11 11.30 56.47 0.46 17.82 12.17 59.03 0.00 0.000

 Contra-Ipsi Lateral Rotation 13.60 9.13 60.25 2.03 13.69 10.75 66.07 0.54 0.919

Upper Lumbar Posterior-Anterior Tilt 42.05 11.84 66.67 9.69 41.35 11.09 66.67 16.63 0.422

 Ipsi-Contra Lateral Tilt 13.77 10.86 64.38 0.00 13.65 10.10 59.13 0.18 0.867

 Contra-Ipsi Lateral Rotation 10.87 8.21 64.26 1.05 11.42 9.24 66.67 0.41 0.456

Lower Lumbar Posterior-Anterior Tilt 37.29 12.85 66.67 11.15 37.79 12.95 66.67 12.54 0.574

 Ipsi-Contra Lateral Tilt 14.82 10.16 55.32 0.00 15.90 11.83 59.77 0.59 0.163

 Contra-Ipsi Lateral Rotation 14.51 9.99 56.92 0.05 16.82 11.77 63.86 0.75 0.002

Pelvis Posterior-Anterior Tilt 38.10 14.02 66.67 6.40 37.77 13.83 66.67 4.94 0.770

 Ipsi-Contra Lateral Tilt 9.83 10.20 65.57 0.28 10.49 11.56 62.98 0.05 0.474

 Contra-Ipsi Lateral Rotation 12.25 8.51 50.01 0.67 11.88 8.95 59.82 0.29 0.602

Hip Flexion-Extension 7.98 10.13 65.29 0.15 8.18 8.86 60.56 0.33 0.787

 Adduction-Abduction 15.39 15.10 66.67 0.84 16.40 14.69 66.62 1.59 0.372

 Internal-External Rotation 28.46 15.05 66.06 3.25 29.51 13.59 63.38 5.99 0.164

Knee Flexion-Extension 6.62 6.64 61.39 0.63 7.71 6.19 62.62 0.40 0.045

 Adduction-Abduction 29.80 16.00 66.05 2.38 30.33 13.42 63.55 2.61 0.545

 Internal-External Rotation 30.06 13.17 62.85 6.59 27.71 12.44 62.08 4.34 0.009

Ankle Dorsi-Plantar Flexion 11.77 9.08 66.67 0.64 10.91 7.88 58.57 1.045 0.294

 Eversion-Inversion 21.22 10.39 57.16 1.87 22.97 11.76 63.11 1.98 0.036

 Internal-External Rotation 30.00 14.85 66.61 1.17 31.13 14.18 66.67 1.22 0.261

Abbreviations: SD, standard deviation; Max, maximum; Min, minimum.

Line 374-394:

Previous studies have investigated spine and lower body gait kinematics [6, 8, 9, 24-27]. In addition, although several studies have investigated on gait symmetry [17-23, 45] and presented valuable information, there is not generally accepted standard for characterization of gait symmetry [23]. Asymmetric gait patterns in healthy individuals may be expected as there exist natural functional differences between the lower extremities [12, 41, 43], such as the contribution of each limb in carrying out the tasks of propulsion and control during able-bodied walking [43]. The present study provides information not only on the degree of asymmetry, the mean angular difference between left and right sides, but also on the SInorm in healthy individuals during normal and fast treadmill walking. Such information will add to the knowledge provided by previous investigations to better understand spine, pelvis, and lower body motions in healthy individuals. Our findings on SInorm for lower body motions in the sagittal plane were comparable to the ones described in [23]. Moreover, this study adds information on the SInorm parameter by describing the spine and pelvis 3D angular motions. In addition, reference degree of asymmetry information in healthy individuals has been presented in this study to help in the biomechanical assessment pathological individuals. Although the use of this indicator may be confusing as it is not referenced to the joint range of motion, such indicator has been implemented to assess asymmetry in pathological individuals. For instance, the degrees of asymmetry reported in total hip [16, 28, 29] and knee [15] replacement patients are greater than the degrees of asymmetry observed in the present study. Consequently, degrees of asymmetry greater than the ones reported in this study may be an indicative of abnormal gait function.

Specific comments:

Line 36 – ‘compared for treadmill walking’ – suggest adding in detail about normal and fast speeds

Reply: We agree and thank the Reviewer for the comment. The sentence has been revised in the Abstract accordingly in order to increase clarity for readership.

Line 36-38:

The spine and pelvis angular motions associated with the left and right lower limb motions, as well as the left and right lower extremity joint angles were compared for normal and fast treadmill walking

Line 67 – ‘common in clinical and rehabilitation practices’ – yes, sure, but these would be for injured or clinical populations, so why not just use overground walking for your healthy adults in this study? Or both overground and treadmill walking?

Reply: We agree and thank the Reviewer for the comment. The goal of this study is to provide reference information on symmetry using healthy individuals, so it can be used for treating injured and clinical populations. Therefore, in this study, we used treadmill walking to generate such reference information as treadmills are commonly used in clinical and rehabilitation practices. 

Line 75-78:

Walking overground is more natural than walking on a treadmill [30-32]. However, the use of treadmills is very common in clinical and rehabilitation practices as they allow for smaller space, better control of walking speeds, and a more controlled environment for kinematics and kinetic studies [32, 33].

Line 73 – this hypothesis seems to be worded as if it is the null hypothesis?

Reply: We agree and thank the Reviewer for the comment. The sentence has been revised in the Introduction Section accordingly in order to increase clarity for readership.

Line 84-88:

We hypothesized that there are significant differences between the spine and pelvis angular movements associated with the left lower limb motions and spine and pelvis angular movements associated with the right lower limb motions, as well as between the left and right lower extremity joint angles during normal and fast treadmill walking in healthy individuals.

Line 82 – what is meant by ‘right-leg dominant’? How was leg dominance defined? Is this relevant here?

Reply: We thank the Reviewer for the comment. Leg dominance was defined as the preferred leg for kicking a ball. We have made clear in the Methods Section that fifty-two participants reported to be right-dominant. This has been included as a study limitation as no action was taken to control the dominant leg during the experiment.

Line 94-95:

Fifty-two out of the sixty participants reported to be right-leg dominant (with leg dominance being defined as the preferred leg for kicking a ball).

Line 394-397:

limitations need to be considered to interpret the present results. To begin with, the average age of the male and female participants in this study was ~21 years old, 52 out of 60 participants reported to be right-dominant, and all participants reported a healthy lifestyle (exercised at least twice a week); hence, results may be limited to similar populations.

Line 91-93 – I’m assuming the walking speeds were determined in separate trials (e.g. 4 x regular; 4 x fast) and not a within-trial acceleration?

Reply: We thank the Reviewer for the comment. Yes, the walking speeds were determined in separate trials. We have made this clear in the Methods Section in order to increase the clarity for readership.

Line 103-107:

All participants first executed normal and fast level overground walking over a distance of 5 m for 4 times in separate trials. Participants were instructed to sustain a usual regular pace during normal overground walking, and accelerate their usual regular pace (as if they were in a hurry) during fast overground walking. The walking speeds of both conditions were recorded and used to set up the treadmill speeds.

Line 113 – ‘z axes’ – it would be helpful to state explicitly which direction this is (vertical)

Reply: We thank the Reviewer for the comment. Although the z-axes mostly point in the vertical direction during standing and treadmill walking, those axes are the local axes attached to spine segments. Therefore, they are not exactly in the vertical direction as they change with participants’ anatomy and motion.

Line 126-130 – were kinematics filtered at all?

Reply: We thank the Reviewer for the comment. Although no filter was applied to the raw data, all kinematics data were indirectly filtered, as the average of at least 30 complete cycles was used for analysis. 

Line 113-116:

Each participant performed three trials that included at least ten complete gait cycles at normal and fast walking speeds. Thus, in total, each test condition had at least 30 complete gait cycles, and those were selected for analyses.

Statistical analyses – (i) It appears that males and females were combined. Would there be reason to separate these two groups to evaluate for sex-based characteristics/differences? (ii) Why weren’t statistical analyses included to compare the treadmill walking speeds?

Reply: We thank the Reviewer for the comment. It is possible to analyze sex-based characteristics/differences. However, the present study has mainly focused on the analysis of gait symmetry for males and females together. 

Furthermore, the information from the normalized symmetry index has been introduced to statistically analyze the differences between normal and fast treadmill walking.

Line 333-335:

Descriptive statistics of the SInorm and its comparison between normal and fast treadmill walking is presented in Table 3. Overall, greater asymmetries were found during fast treadmill walking than normal treadmill walking.

Table 3. Descriptive statistics of the maximum SInorm in % and its comparison between normal and fast treadmill walking for N=60 participants.

Segment Motion Treadmill Walking Normal Treadmill Walking Fast 

 Mean SD Max Min Mean SD Max Min p-value

Upper Thorax Posterior-Anterior Tilt 45.56 12.47 66.67 12.70 44.17 12.61 65.61 12.64 0.201

 Ipsi-Contra Lateral Tilt 13.87 11.07 60.71 0.56 15.36 12.07 63.82 0.84 0.109

 Contra-Ipsi Lateral Rotation 22.28 14.08 64.70 0.81 22.59 12.30 58.93 1.28 0.765

Lower Thorax Posterior-Anterior Tilt 44.88 11.76 66.50 15.60 42.42 11.28 65.84 9.50 0.019

 Ipsi-Contra Lateral Tilt 15.11 11.30 56.47 0.46 17.82 12.17 59.03 0.00 0.000

 Contra-Ipsi Lateral Rotation 13.60 9.13 60.25 2.03 13.69 10.75 66.07 0.54 0.919

Upper Lumbar Posterior-Anterior Tilt 42.05 11.84 66.67 9.69 41.35 11.09 66.67 16.63 0.422

 Ipsi-Contra Lateral Tilt 13.77 10.86 64.38 0.00 13.65 10.10 59.13 0.18 0.867

 Contra-Ipsi Lateral Rotation 10.87 8.21 64.26 1.05 11.42 9.24 66.67 0.41 0.456

Lower Lumbar Posterior-Anterior Tilt 37.29 12.85 66.67 11.15 37.79 12.95 66.67 12.54 0.574

 Ipsi-Contra Lateral Tilt 14.82 10.16 55.32 0.00 15.90 11.83 59.77 0.59 0.163

 Contra-Ipsi Lateral Rotation 14.51 9.99 56.92 0.05 16.82 11.77 63.86 0.75 0.002

Pelvis Posterior-Anterior Tilt 38.10 14.02 66.67 6.40 37.77 13.83 66.67 4.94 0.770

 Ipsi-Contra Lateral Tilt 9.83 10.20 65.57 0.28 10.49 11.56 62.98 0.05 0.474

 Contra-Ipsi Lateral Rotation 12.25 8.51 50.01 0.67 11.88 8.95 59.82 0.29 0.602

Hip Flexion-Extension 7.98 10.13 65.29 0.15 8.18 8.86 60.56 0.33 0.787

 Adduction-Abduction 15.39 15.10 66.67 0.84 16.40 14.69 66.62 1.59 0.372

 Internal-External Rotation 28.46 15.05 66.06 3.25 29.51 13.59 63.38 5.99 0.164

Knee Flexion-Extension 6.62 6.64 61.39 0.63 7.71 6.19 62.62 0.40 0.045

 Adduction-Abduction 29.80 16.00 66.05 2.38 30.33 13.42 63.55 2.61 0.545

 Internal-External Rotation 30.06 13.17 62.85 6.59 27.71 12.44 62.08 4.34 0.009

Ankle Dorsi-Plantar Flexion 11.77 9.08 66.67 0.64 10.91 7.88 58.57 1.045 0.294

 Eversion-Inversion 21.22 10.39 57.16 1.87 22.97 11.76 63.11 1.98 0.036

 Internal-External Rotation 30.00 14.85 66.61 1.17 31.13 14.18 66.67 1.22 0.261

Abbreviations: SD, standard deviation; Max, maximum; Min, minimum.

Line 171 – the values reported here (and throughout the rest of the results section) for detected degrees of asymmetry are extremely small (e.g. posterior tilt 0.46+/- 0.03 degree) and in many cases for a very short period of time (e.g. lower lumbar p/a tilt at normal speed – 2-3% gait cycle). Could they be prone to marker placement error? Or artefacts from the SPM analysis (e.g. as a result of temporal shifting – I don’t think this is the case, but more details on how SPM was performed in the methods section might also help)? With that in mind, are these results meaningful?

Reply: We agree and thank the Reviewer for the comment. The degree of asymmetry is small as it is not referenced to the different ranges of motions of different movements. Yet, it is a parameter that has been reported in the literature. In addition to the degree of asymmetry, the normalized symmetry index has been calculated and discussed accordingly in order to increase the clarity for readership.

Small variations may not be prone to marker placement error as one experienced researcher consistently placed the markers on the bony anatomical landmarks of all participants in this study. 

More details on SPM analysis have been introduced accordingly in the Results Section in order to increase the clarity for readership. The Results Section has been revised and the SPM threshold has been included accordingly. The SPM threshold is calculated automatically by using the SPM 1D function in MATALB. The threshold value must be exceeded to detect significant differences. Fig. 4 illustrates examples of thresholds t* for different motions.

The degree of asymmetry and the normalized symmetry index have been reported in the literature. Thus, we have included and discussed this information in order to make the paper clearer.

Line 124-128:

Average and standard deviation of upper thorax, lower thorax, upper lumbar, and lower lumbar posterior-anterior (P/A) tilt, ipsi-contralateral (Ipsi/Con) tilt, and contra-ipsi (Con/Ipsi) rotation, for left and right sides during one gait cycle of fast treadmill walking (TWF) in sixty healthy participants. Green bars on the horizontal axis and the scalar field SPM results with threshold t* depict where, in % cycle, left side angles were greater or lesser than right side angles. The normalized symmetry index (SInorm) calculated during one gait cycle of TWF. Solid and dashed lines correspond to average left and right sides, as well as average SInorm, and shaded areas correspond to standard deviation. Black dotted vertical lines denote toe-off.

Figure 4:

Line 242-245:

Descriptive statistics of the average degree of asymmetry, describing the mean difference between left and right-side movements when the scalar field SPM detected significant differences, and the maximum magnitude of the SInorm when the scalar field SPM detected significant differences, in spine segments during normal and fast treadmill walking are presented in Table 1.

Line 321-325

Descriptive statistics of the average degree of asymmetry, describing the mean difference between left and right-side movements when the scalar field SPM detected significant differences, and the maximum magnitude of the SInorm when the scalar field SPM detected significant differences, in the pelvis segment and lower body joints during normal and fast treadmill walking are presented in Table 2.

Line 333-335:

Descriptive statistics of the SInorm and its comparison between normal and fast treadmill walking is presented in Table 3. Overall, greater asymmetries were found during fast treadmill walking than normal treadmill walking.

Line 374-394:

Previous studies have investigated spine and lower body gait kinematics [6, 8, 9, 24-27]. In addition, although several studies have investigated on gait symmetry [17-23, 45] and presented valuable information, there is not generally accepted standard for characterization of gait symmetry [23]. Asymmetric gait patterns in healthy individuals may be expected as there exist natural functional differences between the lower extremities [12, 41, 43], such as the contribution of each limb in carrying out the tasks of propulsion and control during able-bodied walking [43]. The present study provides information not only on the degree of asymmetry, the mean angular difference between left and right sides, but also on the SInorm in healthy individuals during normal and fast treadmill walking. Such information will add to the knowledge provided by previous investigations to better understand spine, pelvis, and lower body motions in healthy individuals. Our findings on SInorm for lower body motions in the sagittal plane were comparable to the ones described in [23]. Moreover, this study adds information on the SInorm parameter by describing the spine and pelvis 3D angular motions. In addition, reference degree of asymmetry information in healthy individuals has been presented in this study to help in the biomechanical assessment pathological individuals. Although the use of this indicator may be confusing as it is not referenced to the joint range of motion, such indicator has been implemented to assess asymmetry in pathological individuals. For instance, the degrees of asymmetry reported in total hip [16, 28, 29] and knee [15] replacement patients are greater than the degrees of asymmetry observed in the present study. Consequently, degrees of asymmetry greater than the ones reported in this study may be an indicative of abnormal gait function.

Line 300 – ‘internal notation’ – I think this is meant to be ‘rotation’?

Reply: We agree and thank the Reviewer for the comment. The sentence has been revised in the Introduction and Discussion Sections accordingly in order to increase clarity for readership.

Line 349-352:

Degrees of asymmetry and the associated maximum magnitude of SInorm of 5.74±0.04° and 14%, as well as 5.33±0.18° and 21%, for the lower lumbar contra-ipsi lateral rotation and hip internal rotation, respectively, were the largest asymmetries detected during normal treadmill walking.

Line 305 – links to comment above regarding hypothesis – this statement seems to conflict with the way the hypothesis was worded (original hypothesis suggests there would not be any differences, but you have found some, so null hypothesis (that there would be differences) is, at least, partially supported here).

Reply: We agree and thank the Reviewer for the comment. The sentence has been revised in the Introduction Section accordingly in order to increase clarity for readership.

Line 84-88:

We hypothesized that there are significant differences between the spine and pelvis angular movements associated with the left lower limb motions and spine and pelvis angular movements associated with the right lower limb motions, as well as between the left and right lower extremity joint angles during normal and fast treadmill walking in healthy individuals.

Line 357-359:

These results rejected the null hypothesis of no difference in spine, pelvis, and lower body motions between left and right sides during normal and fast treadmill walking in healthy individuals.

Line 312 – again, why didn’t you make these comparisons (i.e. through statistical testing)?

Reply: We thank the Reviewer for the comment. The information from the normalized symmetry index has been introduced to statistically analyze the differences between normal and fast treadmill walking. This has been discussed accordingly in order to increase the clarity for readership.

Line 333-335:

Descriptive statistics of the SInorm and its comparison between normal and fast treadmill walking is presented in Table 3. Overall, greater asymmetries were found during fast treadmill walking than normal treadmill walking.

Table 3. Descriptive statistics of the maximum SInorm in % and its comparison between normal and fast treadmill walking for N=60 participants.

Segment Motion Treadmill Walking Normal Treadmill Walking Fast 

 Mean SD Max Min Mean SD Max Min p-value

Upper Thorax Posterior-Anterior Tilt 45.56 12.47 66.67 12.70 44.17 12.61 65.61 12.64 0.201

 Ipsi-Contra Lateral Tilt 13.87 11.07 60.71 0.56 15.36 12.07 63.82 0.84 0.109

 Contra-Ipsi Lateral Rotation 22.28 14.08 64.70 0.81 22.59 12.30 58.93 1.28 0.765

Lower Thorax Posterior-Anterior Tilt 44.88 11.76 66.50 15.60 42.42 11.28 65.84 9.50 0.019

 Ipsi-Contra Lateral Tilt 15.11 11.30 56.47 0.46 17.82 12.17 59.03 0.00 0.000

 Contra-Ipsi Lateral Rotation 13.60 9.13 60.25 2.03 13.69 10.75 66.07 0.54 0.919

Upper Lumbar Posterior-Anterior Tilt 42.05 11.84 66.67 9.69 41.35 11.09 66.67 16.63 0.422

 Ipsi-Contra Lateral Tilt 13.77 10.86 64.38 0.00 13.65 10.10 59.13 0.18 0.867

 Contra-Ipsi Lateral Rotation 10.87 8.21 64.26 1.05 11.42 9.24 66.67 0.41 0.456

Lower Lumbar Posterior-Anterior Tilt 37.29 12.85 66.67 11.15 37.79 12.95 66.67 12.54 0.574

 Ipsi-Contra Lateral Tilt 14.82 10.16 55.32 0.00 15.90 11.83 59.77 0.59 0.163

 Contra-Ipsi Lateral Rotation 14.51 9.99 56.92 0.05 16.82 11.77 63.86 0.75 0.002

Pelvis Posterior-Anterior Tilt 38.10 14.02 66.67 6.40 37.77 13.83 66.67 4.94 0.770

 Ipsi-Contra Lateral Tilt 9.83 10.20 65.57 0.28 10.49 11.56 62.98 0.05 0.474

 Contra-Ipsi Lateral Rotation 12.25 8.51 50.01 0.67 11.88 8.95 59.82 0.29 0.602

Hip Flexion-Extension 7.98 10.13 65.29 0.15 8.18 8.86 60.56 0.33 0.787

 Adduction-Abduction 15.39 15.10 66.67 0.84 16.40 14.69 66.62 1.59 0.372

 Internal-External Rotation 28.46 15.05 66.06 3.25 29.51 13.59 63.38 5.99 0.164

Knee Flexion-Extension 6.62 6.64 61.39 0.63 7.71 6.19 62.62 0.40 0.045

 Adduction-Abduction 29.80 16.00 66.05 2.38 30.33 13.42 63.55 2.61 0.545

 Internal-External Rotation 30.06 13.17 62.85 6.59 27.71 12.44 62.08 4.34 0.009

Ankle Dorsi-Plantar Flexion 11.77 9.08 66.67 0.64 10.91 7.88 58.57 1.045 0.294

 Eversion-Inversion 21.22 10.39 57.16 1.87 22.97 11.76 63.11 1.98 0.036

 Internal-External Rotation 30.00 14.85 66.61 1.17 31.13 14.18 66.67 1.22 0.261

Abbreviations: SD, standard deviation; Max, maximum; Min, minimum.

Line 67-74:

Furthermore, a classical 

Line 321 – some examples of the differences to which you refer here would be helpful

Reply: We thank the Reviewer for the comment. Examples of the natural functional differences between the lower extremities has been included to make the paper clearer.

Line 360-366:

It has been reported that the walking speed affects individuals’ gait kinematics [39, 40]; however, the influence of treadmill walking speed on gait symmetry kinematics in healthy individuals remains unclear. Even though it has been reported that the imposed constant speed of a treadmill may artificially impose motor control of gait and impede the natural variation that occurs during overground walking and therefore minimize gait variability [41, 42], overall, our findings suggest that young healthy adults may be more asymmetrical during fast treadmill walking than normal treadmill walking (see Table 3).

Line 333 – the wording of this limitation isn’t clear to me; what do you mean by ‘few gait cycles with repetitions’?

Reply: We thank the Reviewer for the comment. We have clarified in the Limitations Section that few gait cycles (~30) were used for were used in normal and fast treadmill walking conditions. 

Line 398-399:

Furthermore, few gait cycles (~30) were used in normal and fast treadmill walking conditions; hence, the long terms of gait asymmetry kinematics were not explored.

Line 345 – again, you didn’t run any tests so I think this statement needs to address that.

Reply: We thank the Reviewer for the comment. The information from the normalized symmetry index has been introduced to statistically analyze the differences between normal and fast treadmill walking. This has been discussed accordingly in order to increase the clarity for readership.

Line 67-74:

Furthermore, a classical 

Line 347 – as per my main comment with the discussion section, a summary statement of the significance of these very small degrees of asymmetry should be included. What is the take-home message/impact of these findings?

Reply: We agree and thank the Reviewer for the comment. The degree of asymmetry is small as it is not referenced to the different range of motions of different movements. Yet, it is a parameter that has been reported in the literature. In addition to the degree of asymmetry, the normalized symmetry index has been calculated and discussed accordingly in order to increase the clarity for readership.

Line 333-335:

Descriptive statistics of the SInorm and its comparison between normal and fast treadmill walking is presented in Table 3. Overall, greater asymmetries were found during fast treadmill walking than normal treadmill walking.

Table 3. Descriptive statistics of the maximum SInorm in % and its comparison between normal and fast treadmill walking for N=60 participants.

Segment Motion Treadmill Walking Normal Treadmill Walking Fast 

 Mean SD Max Min Mean SD Max Min p-value

Upper Thorax Posterior-Anterior Tilt 45.56 12.47 66.67 12.70 44.17 12.61 65.61 12.64 0.201

 Ipsi-Contra Lateral Tilt 13.87 11.07 60.71 0.56 15.36 12.07 63.82 0.84 0.109

 Contra-Ipsi Lateral Rotation 22.28 14.08 64.70 0.81 22.59 12.30 58.93 1.28 0.765

Lower Thorax Posterior-Anterior Tilt 44.88 11.76 66.50 15.60 42.42 11.28 65.84 9.50 0.019

 Ipsi-Contra Lateral Tilt 15.11 11.30 56.47 0.46 17.82 12.17 59.03 0.00 0.000

 Contra-Ipsi Lateral Rotation 13.60 9.13 60.25 2.03 13.69 10.75 66.07 0.54 0.919

Upper Lumbar Posterior-Anterior Tilt 42.05 11.84 66.67 9.69 41.35 11.09 66.67 16.63 0.422

 Ipsi-Contra Lateral Tilt 13.77 10.86 64.38 0.00 13.65 10.10 59.13 0.18 0.867

 Contra-Ipsi Lateral Rotation 10.87 8.21 64.26 1.05 11.42 9.24 66.67 0.41 0.456

Lower Lumbar Posterior-Anterior Tilt 37.29 12.85 66.67 11.15 37.79 12.95 66.67 12.54 0.574

 Ipsi-Contra Lateral Tilt 14.82 10.16 55.32 0.00 15.90 11.83 59.77 0.59 0.163

 Contra-Ipsi Lateral Rotation 14.51 9.99 56.92 0.05 16.82 11.77 63.86 0.75 0.002

Pelvis Posterior-Anterior Tilt 38.10 14.02 66.67 6.40 37.77 13.83 66.67 4.94 0.770

 Ipsi-Contra Lateral Tilt 9.83 10.20 65.57 0.28 10.49 11.56 62.98 0.05 0.474

 Contra-Ipsi Lateral Rotation 12.25 8.51 50.01 0.67 11.88 8.95 59.82 0.29 0.602

Hip Flexion-Extension 7.98 10.13 65.29 0.15 8.18 8.86 60.56 0.33 0.787

 Adduction-Abduction 15.39 15.10 66.67 0.84 16.40 14.69 66.62 1.59 0.372

 Internal-External Rotation 28.46 15.05 66.06 3.25 29.51 13.59 63.38 5.99 0.164

Knee Flexion-Extension 6.62 6.64 61.39 0.63 7.71 6.19 62.62 0.40 0.045

 Adduction-Abduction 29.80 16.00 66.05 2.38 30.33 13.42 63.55 2.61 0.545

 Internal-External Rotation 30.06 13.17 62.85 6.59 27.71 12.44 62.08 4.34 0.009

Ankle Dorsi-Plantar Flexion 11.77 9.08 66.67 0.64 10.91 7.88 58.57 1.045 0.294

 Eversion-Inversion 21.22 10.39 57.16 1.87 22.97 11.76 63.11 1.98 0.036

 Internal-External Rotation 30.00 14.85 66.61 1.17 31.13 14.18 66.67 1.22 0.261

Abbreviations: SD, standard deviation; Max, maximum; Min, minimum.

Line 412-414:

Our findings suggest that young healthy adults may be more asymmetrical during fast treadmill walking than normal treadmill walking. 

Figure 3 – legend for Upper Lumbar P/A tilt has ‘GWN’ which isn’t explained anywhere. I’m assuming it is ‘ground walking normal’, but might be a typo?

Reply: We thank the Reviewer for the comment. All figures have been revised and corrected accordingly in order to increase the clarity for readership.

Line 209-216:

Fig 3. Average and standard deviation of upper thorax, lower thorax, upper lumbar, and lower lumbar posterior-anterior (P/A) tilt, ipsi-contralateral (Ipsi/Con) tilt, and contra-ipsi (Con/Ipsi) rotation, for left and right sides during one gait cycle of normal treadmill walking (TWN) in sixty healthy participants. Green bars on the horizontal axis and the scalar field SPM results with threshold t* depict where, in % cycle, left side angles were greater or lesser than right side angles. The normalized symmetry index (SInorm) calculated during one gait cycle of TWN. Solid and dashed lines correspond to average left and right sides, as well as average SInorm, and shaded areas correspond to standard deviation. Black dotted vertical lines denote toe-off.

Figure 3:

---

## [Decision Letter · Decision Letter 1]

4 Sep 2022

PONE-D-22-08871R1

Spine and lower body symmetry during treadmill walking in healthy individuals - in-vivo 3-dimensional kinematic analysis

PLOS ONE

Dear Dr. Arauz,

Thank you for submitting your manuscript to PLOS ONE. After careful consideration, we feel that it has merit but does not fully meet PLOS ONE’s publication criteria as it currently stands. Therefore, we invite you to submit a revised version of the manuscript that addresses the points raised during the review process.

We look forward to receiving your revised manuscript.

Kind regards,

Theodoros M. Bampouras

Academic Editor

PLOS ONE

Journal Requirements:

Additional Editor Comments:

I have now received reviewers' comments from the required number of reviewers. You will see from the reviewers' comments that there are some minor outstanding matters that need to be addressed, before the manuscript can be accepted for publication; mostly around clarification of terms and concepts. I encourage you to consider and address them carefully in your reply, as per the instructions below. 

Reviewers' comments:

Reviewer's Responses to Questions

**Comments to the Author**

1. If the authors have adequately addressed your comments raised in a previous round of review and you feel that this manuscript is now acceptable for publication, you may indicate that here to bypass the “Comments to the Author” section, enter your conflict of interest statement in the “Confidential to Editor” section, and submit your "Accept" recommendation.

Reviewer #3: All comments have been addressed

Reviewer #4: (No Response)

2. Is the manuscript technically sound, and do the data support the conclusions?

Reviewer #3: Yes

Reviewer #4: Yes

3. Has the statistical analysis been performed appropriately and rigorously? 

Reviewer #3: Yes

Reviewer #4: Yes

4. Have the authors made all data underlying the findings in their manuscript fully available?

Reviewer #3: Yes

Reviewer #4: Yes

5. Is the manuscript presented in an intelligible fashion and written in standard English?

Reviewer #3: Yes

Reviewer #4: Yes

6. Review Comments to the Author

Reviewer #3: This is an interesting paper on spinal and lower body symmetry during treadmill walking. The authors have addressed all previous comments and the paper flows significanlty better. Large sections of the results and discussion section have been revised in order to increase clarity for readership.

Reviewer #4: Title: Spine and lower body symmetry during treadmill walking in healthy individuals – in-vivo 3-dimensional kinematic analysis

The manuscript describes a cross-sectional observational study investigating the bilateral symmetry of 3-dimensional motions of the spine (four segments), pelvis, and lower limb joints during treadmill walking at an individual’s regular walking speed and fast walking speed (i.e., “as if they were in a hurry”). Overall, the manuscript is relatively well-written and clear, although a number of minor grammatical errors were noted in the document. Below are substantive comments for the authors to consider in their revisions.

Terminology

The terminology used to describe the 3-dimensional motions of the spine and pelvis segments are confusing.

Regarding spinal motion:

- “flexion/extension” is typically used for sagittal plane motion (rather than “posterior-anterior tilt”)

- “lateral flexion” is typically used for frontal plane motion (rather than “lateral tilt”)

- “(left/right) rotation” is typically used for transverse plane motion (rather than “lateral rotation”)

Regarding pelvis motion:

- “obliquity” is typically used for frontal plane motion (rather than “lateral tilt”)

- “(left/right) rotation” is typically used for transverse plane motion (rather than “lateral rotation”)

As well, the terms “ipsi-contra” or “contra-ipsi” are frequently used along with the motion terms – e.g., lower lumbar contra-ipsi lateral rotation (Line 38), upper lumbar ipsi-contra lateral tilt (Line 40). It is unclear what these represent – e.g., does “lower lumbar contra-ipsi lateral rotation” mean transverse plane rotation of the lower lumbar segment from the side contralateral to the leg being analyzed to the side ipsilateral to that leg?

It would be very helpful to use the ISB’s current guidelines regarding the terminology to use when describing the three-dimensional motions of the spine and pelvis segments to avoid confusion for the readers of this manuscript. Such changes would need to be made throughout the manuscript (e.g., Abstract, Results, Discussion, Tables, Figures).

Methods (Participants)

Was an a priori sample size estimate/power analysis performed? If so, please provide the appropriate details regarding this estimate/analysis in the manuscript. If not, please provide a rationale for not doing so in the manuscript

Methods (Procedures and data collection)

Please state the type of Vicon cameras that were used (e.g., Vantage) and their resolution in the manuscript.

Figure 1 (Legend)

The abbreviation “L” is used twice (for “Lower” and “Lateral”). Please change one of the abbreviations to avoid confusion.

Discussion

- Lines 362-366, 372-373: In the clinical and research literature, the term “gait variability” refers to the stride-to-stride fluctuations in a gait parameter (e.g., step time/length, stride time/length). This is very different than the construct assessed in this study (i.e., “symmetry” as defined in the manuscript). Therefore, these sentences are not appropriate and need to be edited for clarity.

- Lines 367-368: This sentence alludes to assessments made “during normal and treadmill walking”. However, no assessments were made using data collected during normal overground walking. Therefore, this sentence is not appropriate and needs to be edited for clarity.

- Lines 370-372: This sentence alludes to assessments made using data collected while individuals were “walking slowly”. However, no such assessments were made (asking individuals to walk at their regular walking speed is not the same as asking them to walk at a pace slower than their regular walking speed). Therefore, this sentence is not appropriate and needs to be edited for clarity.

7. PLOS authors have the option to publish the peer review history of their article (what does this mean?). If published, this will include your full peer review and any attached files.

Reviewer #3: **Yes: **Dr Konstantinos Papadopoulos

Reviewer #4: No

---

## [Author Response · Author response to Decision Letter 1]

6 Sep 2022

AUTHOR’S RESPONSE TO REVIEWER’S COMMENTS 9/6/2022

The authors wish to thank the Editor and Reviewers for their invaluable suggestions. The manuscript has been revised according to the editor’s and reviewers’ comments. A point-by-point reply is given below. The line, figure and table numbers noted in the replies refer to the revised manuscript. In the following response, the editor’s and reviewer’s comments are shown in italics and our responses appear immediately below in normal type.

Editor:

Additional Editor Comments:

I have now received reviewers' comments from the required number of reviewers. You will see from the reviewers' comments that there are some minor outstanding matters that need to be addressed, before the manuscript can be accepted for publication; mostly around clarification of terms and concepts. I encourage you to consider and address them carefully in your reply, as per the instructions below.

Reply: We thank the Editor for the comments. All comments have been addressed accordingly to increase the clarity for readership.

Reviewer #3

Reviewer #3: This is an interesting paper on spinal and lower body symmetry during treadmill walking. The authors have addressed all previous comments and the paper flows significantly better. Large sections of the results and discussion section have been revised in order to increase clarity for readership.

Reply: We thank the Reviewer for the comments. 

Reviewer #4

Reviewer #4: Title: Spine and lower body symmetry during treadmill walking in healthy individuals – in-vivo 3-dimensional kinematic analysis

The manuscript describes a cross-sectional observational study investigating the bilateral symmetry of 3-dimensional motions of the spine (four segments), pelvis, and lower limb joints during treadmill walking at an individual’s regular walking speed and fast walking speed (i.e., “as if they were in a hurry”). Overall, the manuscript is relatively well-written and clear, although a number of minor grammatical errors were noted in the document. Below are substantive comments for the authors to consider in their revisions.

Reply: We thank the Reviewer for the comments. We have addressed all comments accordingly to increase the clarity for readership.

Terminology

The terminology used to describe the 3-dimensional motions of the spine and pelvis segments are confusing.

Regarding spinal motion:

- “flexion/extension” is typically used for sagittal plane motion (rather than “posterior-anterior tilt”)

- “lateral flexion” is typically used for frontal plane motion (rather than “lateral tilt”)

- “(left/right) rotation” is typically used for transverse plane motion (rather than “lateral rotation”)

Regarding pelvis motion:

- “obliquity” is typically used for frontal plane motion (rather than “lateral tilt”)

- “(left/right) rotation” is typically used for transverse plane motion (rather than “lateral rotation”)

As well, the terms “ipsi-contra” or “contra-ipsi” are frequently used along with the motion terms – e.g., lower lumbar contra-ipsi lateral rotation (Line 38), upper lumbar ipsi-contra lateral tilt (Line 40). It is unclear what these represent – e.g., does “lower lumbar contra-ipsi lateral rotation” mean transverse plane rotation of the lower lumbar segment from the side contralateral to the leg being analyzed to the side ipsilateral to that leg?

It would be very helpful to use the ISB’s current guidelines regarding the terminology to use when describing the three-dimensional motions of the spine and pelvis segments to avoid confusion for the readers of this manuscript. Such changes would need to be made throughout the manuscript (e.g., Abstract, Results, Discussion, Tables, Figures).

Reply: We agree and thank the Reviewer for the comments. The terminology has been revised throughout all Sections, as well as Figures and Tables of the manuscript to increase the clarity for readership.

Line 38-41:

The lower lumbar left-right rotation (5.74±0.04°) and hip internal rotation (5.33±0.18°) presented the largest degrees of asymmetry during normal treadmill. Upper lumbar left-right lateral flexion (1.48±0.14°) and knee flexion (2.98±0.13°) indicated the largest asymmetries and during fast treadmill walking.

Figure 3.

Figure 4.

Figure 5.

Figure 6.

Table 1. Average and standard deviation of the degree of asymmetry and maximum magnitude of the normalized symmetry index between the associated left and right motions in spine segments during normal and fast treadmill walking of N=60 participants. 

Segment Treadmill Walking Normal Treadmill Walking Fast

Upper Lumbar Flexion-Extension Flexion-Extension 

 Gait Cycle % DoA p-value max SInorm % Gait Cycle % DoA p-value max SInorm %

 45 to 52 0.46±0.03 0.003 30 

 93 to 99 0.48±0.05 0.004 30 

 Left-Right Lateral Flexion Left-Right Lateral Flexion 

 Gait Cycle % DoA p-value max SInorm % Gait Cycle % DoA p-value max SInorm %

 0 to 100 1.54±0.11 <0.001 13 0 to 100 1.48±0.14 <0.001 15.72

 Left-Right Rotation Left-Right Rotation 

 Gait Cycle % DoA p-value max SInorm % Gait Cycle % DoA p-value max SInorm %

 13 to 20 0.64±0.016 0.04 11

 60 to 73 0.68±0.054 0.021 13.3

Lower Lumbar Flexion-Extension Flexion-Extension 

 Gait Cycle % DoA p-value max SInorm % Gait Cycle % DoA p-value max SInorm %

 2 to 3 0.48±0.01 0.049 29 9 to 19 0.68±0.05 0.001 30

 4 to 16 0.45±0.09 0.002 32.7 58 to 68 0.51±0.12 0.001 34

 55 to 66 0.42±0.09 0.005 32.7 

 Left-Right Lateral Flexion Left-Right Lateral Flexion 

 Gait Cycle % DoA p-value max SInorm % Gait Cycle % DoA p-value max SInorm %

 1 to 8 1.3±0.02 0.048 9.7 

 14 to 45 1.42±0.06 0.023 14 

 60 to 98 1.42±0.08 0.016 12.5 

 99 to 100 1.32±0.03 0.05 10 

 Left-Right Rotation Left-Right Rotation 

 Gait Cycle % DoA p-value max SInorm % Gait Cycle % DoA p-value max SInorm %

 0 to 100 5.74±0.04 0.006 14.73 

Abbreviations: DoA, degree of asymmetry; SInorm, normalized symmetry index.

Table 2. Average and standard deviation of the degree of asymmetry and maximum magnitude of the normalized symmetry index between the associated left and right motions in the pelvis segment and the lower body joints during normal and fast treadmill walking of N=60 participants. 

Segment Treadmill Walking Normal Treadmill Walking Fast

Pelvis Posterior-Anterior Tilt Posterior-Anterior Tilt 

 Gait Cycle % DoA p-value max SInorm % Gait Cycle % DoA p-value max SInorm %

 30 to 40 0.22±0.01 0.013 26.8 0 to 13 051±0.03 0.004 35.8

 48 to 49 0.26±0.01 0.05 27.6 36 to 63 0.46±0.03 <0.001 37.4

 76 to 93 0.22±0.02 0.001 27.2 84 to 100 0.48±0.05 0.001 24.6

 Left-Right Obliquity Left-Right Obliquity 

 Gait Cycle % DoA p-value max SInorm % Gait Cycle % DoA p-value max SInorm %

 0 to 100 1.38±0.06 0.001 11.5 4 to 22 1.06±0.06° 0.037 10.3

 53 to 66 1.07±0.03° 0.043 11

Hip Flexion-Extension Flexion-Extension 

 Gait Cycle % DoA p-value max SInorm % Gait Cycle % DoA p-value max SInorm %

 15 to 55 2.57±0.08 0.01 11.8 19 to 53 2.60±0.13 0.004 10

 Internal-External Rotation Internal-External Rotation 

 Gait Cycle % DoA p-value max SInorm % Gait Cycle % DoA p-value max SInorm %

 4 to 10 5.33±0.18 0.049 21 

 68 to 78 5.27±0.08 0.048 22 

Knee Flexion-Extension Flexion-Extension 

 Gait Cycle % DoA p-value max SInorm % Gait Cycle % DoA p-value max SInorm %

 5 to 39 2.65±0.1 0.009 7.9 0 to 42 2.81±0.24 0.001 8.3

 82 to 97 2.71±0.13 0.031 5.3 81 to 100 2.98±0.13 0.02 6.5

 Internal-External Rotation Internal-External Rotation 

 Gait Cycle % DoA p-value max SInorm % Gait Cycle % DoA p-value max SInorm %

 4 to 10 1.85±0.12 0.049 18.5 0 to 20 2.35±0.35 <0.001 22.7

 16 to 17 1.64±0.01 0.05 12.6 25 to 40 1.89±0.11 0.001 15.4

 22 to 42 1.75±0.1 0.001 18.5 95 to 100 2.29±0.25 0.031 22.7

 83 to 87 1.57±0.08 0.044 18.5 

 96 to 97 1.75±0.1 0.05 22.7 

Ankle Dorsi-Plantar Flexion Dorsi-Plantar Flexion 

 Gait Cycle % DoA p-value max SInorm % Gait Cycle % DoA p-value max SInorm %

 12 to 13 1.62±0.08 0.05 8.4

 35 to 37 1.21±0.01 0.05 6.5

 45 to 48 1.34±0.16 0.039 7.8

 79 to 94 1.37±0.14 <0.001 8.4

 Eversion-Inversion Eversion-Inversion 

 Gait Cycle % DoA p-value max SInorm % Gait Cycle % DoA p-value max SInorm %

 0 to 19 2.34±0.18 0.021 13.4 

 71 to 100 2.6±0.22 0.008 16.4 

 Internal-External Rotation Internal-External Rotation 

 Gait Cycle % DoA p-value max SInorm % Gait Cycle % DoA p-value max SInorm %

 92 to 98 1.76±0.05 0.034 24.5

Abbreviations: DoA, degree of asymmetry; SInorm, normalized symmetry index.

Table 3. Descriptive statistics of the maximum SInorm in % and its comparison between normal and fast treadmill walking for N=60 participants.

Segment Motion Treadmill Walking Normal Treadmill Walking Fast 

 Mean SD Max Min Mean SD Max Min p-value

Upper Thorax Flexion-Extension 45.56 12.47 66.67 12.70 44.17 12.61 65.61 12.64 0.201

 Left-Right Lateral Flexion 13.87 11.07 60.71 0.56 15.36 12.07 63.82 0.84 0.109

 Left-Right Rotation 22.28 14.08 64.70 0.81 22.59 12.30 58.93 1.28 0.765

Lower Thorax Flexion-Extension 44.88 11.76 66.50 15.60 42.42 11.28 65.84 9.50 0.019

 Left-Right Lateral Flexion 15.11 11.30 56.47 0.46 17.82 12.17 59.03 0.00 0.000

 Left-Right Rotation 13.60 9.13 60.25 2.03 13.69 10.75 66.07 0.54 0.919

Upper Lumbar Flexion-Extension 42.05 11.84 66.67 9.69 41.35 11.09 66.67 16.63 0.422

 Left-Right Lateral Flexion 13.77 10.86 64.38 0.00 13.65 10.10 59.13 0.18 0.867

 Left-Right Rotation 10.87 8.21 64.26 1.05 11.42 9.24 66.67 0.41 0.456

Lower Lumbar Flexion-Extension 37.29 12.85 66.67 11.15 37.79 12.95 66.67 12.54 0.574

 Left-Right Lateral Flexion 14.82 10.16 55.32 0.00 15.90 11.83 59.77 0.59 0.163

 Left-Right Rotation 14.51 9.99 56.92 0.05 16.82 11.77 63.86 0.75 0.002

Pelvis Posterior-Anterior Tilt 38.10 14.02 66.67 6.40 37.77 13.83 66.67 4.94 0.770

 Left-Right Obliquity 9.83 10.20 65.57 0.28 10.49 11.56 62.98 0.05 0.474

 Left-Right Rotation 12.25 8.51 50.01 0.67 11.88 8.95 59.82 0.29 0.602

Hip Flexion-Extension 7.98 10.13 65.29 0.15 8.18 8.86 60.56 0.33 0.787

 Adduction-Abduction 15.39 15.10 66.67 0.84 16.40 14.69 66.62 1.59 0.372

 Internal-External Rotation 28.46 15.05 66.06 3.25 29.51 13.59 63.38 5.99 0.164

Knee Flexion-Extension 6.62 6.64 61.39 0.63 7.71 6.19 62.62 0.40 0.045

 Adduction-Abduction 29.80 16.00 66.05 2.38 30.33 13.42 63.55 2.61 0.545

 Internal-External Rotation 30.06 13.17 62.85 6.59 27.71 12.44 62.08 4.34 0.009

Ankle Dorsi-Plantar Flexion 11.77 9.08 66.67 0.64 10.91 7.88 58.57 1.045 0.294

 Eversion-Inversion 21.22 10.39 57.16 1.87 22.97 11.76 63.11 1.98 0.036

 Internal-External Rotation 30.00 14.85 66.61 1.17 31.13 14.18 66.67 1.22 0.261

Abbreviations: SD, standard deviation; Max, maximum; Min, minimum.

Methods (Participants)

Was an a priori sample size estimate/power analysis performed? If so, please provide the appropriate details regarding this estimate/analysis in the manuscript. If not, please provide a rationale for not doing so in the manuscript

Reply: We agree and thank the Reviewer for the comment. A power analysis has been included in the Methods Section accordingly to make the manuscript clearer. Based on the study results, a sample size of 60 participants with an alpha = 0.05, and a sample ratio = 1, yields a power = 0.82. 

Line 105-106:

A power analysis based on our study results indicates that a sample size of 60 participants with an alpha = 0.05, and a sample ratio = 1, produces a power = 0.82.

Figure 1 (Legend)

The abbreviation “L” is used twice (for “Lower” and “Lateral”). Please change one of the abbreviations to avoid confusion.

Reply: We agree and thank the Reviewer for the comment. One of the abbreviations had been changed as suggested. The abbreviation “Lt” has been used for “Lateral”.

Line 122-127:

Fig 1. Spine and lower body marker set. Prefixes denote the following: L: Left, R: Right, U: Upper, L: Lower, Lt: Lateral, and M: Medial. The following landmarks were used: Spinous process at T1 (T1), spinous process at T6 (T6), spinous process at L1(L1), spinous process at L3 (L3), spinous process at L5 (L5), thorax (TH), lumbar (LB), anterior superior iliac spine (ASI), posterior superior iliac spine (PSI), femur (THI), epicondyle of femur (KN), tibia (TB), malleoli (AK), and foot (FT).

Figure 1.

Discussion

- Lines 362-366, 372-373: In the clinical and research literature, the term “gait variability” refers to the stride-to-stride fluctuations in a gait parameter (e.g., step time/length, stride time/length). This is very different than the construct assessed in this study (i.e., “symmetry” as defined in the manuscript). Therefore, these sentences are not appropriate and need to be edited for clarity.

Reply: We agree and thank the Reviewer for the comment. The sentence has been removed from the Discussion Section to increase the clarity for readership.

Line 366-367:

Overall, our findings suggest that young healthy adults may be more asymmetrical during fast treadmill walking than normal treadmill walking (see Table 3).

- Lines 367-368: This sentence alludes to assessments made “during normal and treadmill walking”. However, no assessments were made using data collected during normal overground walking. Therefore, this sentence is not appropriate and needs to be edited for clarity.

Reply: We agree thank the Reviewer for the comment. The sentence has been corrected in the Discussion Section accordingly to increase the clarity for readership.

Line 367-369:

In addition, our results revealed standard deviations of the left side higher than the right side for hip, knee, and ankle motions during normal and fast treadmill walking.

- Lines 370-372: This sentence alludes to assessments made using data collected while individuals were “walking slowly”. However, no such assessments were made (asking individuals to walk at their regular walking speed is not the same as asking them to walk at a pace slower than their regular walking speed). Therefore, this sentence is not appropriate and needs to be edited for clarity.

Reply: We agree and thank the Reviewer for the comment. We have revised the sentence accordingly in the Discussion Section to increase the clarity for readership. 

Line 371-376:

Even though previous reports indicate that walking slowly is more challenging to the motor control of gait than usual and faster speed walks [41, 44], differences of gait motor control between usual and faster speed walking are not clear. Therefore, the findings of this study reported as the degree of asymmetry and the normalized symmetry index may be useful indicators of the gait motor control at different walking speeds.

---

## [Decision Letter · Decision Letter 2]

12 Sep 2022

Spine and lower body symmetry during treadmill walking in healthy individuals - in-vivo 3-dimensional kinematic analysis

PONE-D-22-08871R2

Dear Dr. Arauz,

We’re pleased to inform you that your manuscript has been judged scientifically suitable for publication and will be formally accepted for publication once it meets all outstanding technical requirements.

Kind regards,

Theodoros M. Bampouras

Academic Editor

PLOS ONE

Additional Editor Comments (optional):

Please address the Reviewer's comments for clarity at the proof stage.  

Reviewers' comments:

Reviewer's Responses to Questions

**Comments to the Author**

1. If the authors have adequately addressed your comments raised in a previous round of review and you feel that this manuscript is now acceptable for publication, you may indicate that here to bypass the “Comments to the Author” section, enter your conflict of interest statement in the “Confidential to Editor” section, and submit your "Accept" recommendation.

Reviewer #4: All comments have been addressed

2. Is the manuscript technically sound, and do the data support the conclusions?

Reviewer #4: Yes

3. Has the statistical analysis been performed appropriately and rigorously? 

Reviewer #4: Yes

4. Have the authors made all data underlying the findings in their manuscript fully available?

Reviewer #4: Yes

5. Is the manuscript presented in an intelligible fashion and written in standard English?

Reviewer #4: Yes

6. Review Comments to the Author

Reviewer #4: The authors have addressed all of my earlier comments. I only have two further comments that can be resolved during the proofing stage.

1) Lines 122-123 (Figure 1 Legend): The abbreviation "L" is still used twice (for "Left" and "Lower"). It would be helpful to use a different abbreviation for one of the words.

2) Line 214 (Figure 3 Legend): The term "tilt" should be removed from this line.

7. PLOS authors have the option to publish the peer review history of their article (what does this mean?). If published, this will include your full peer review and any attached files.

Reviewer #4: No

---

## [Editor Report · Acceptance letter]

16 Sep 2022

PONE-D-22-08871R2 

Spine and lower body symmetry during treadmill walking in healthy individuals - in-vivo 3-dimensional kinematic analysis 

Dear Dr. Arauz:

I'm pleased to inform you that your manuscript has been deemed suitable for publication in PLOS ONE. Congratulations! Your manuscript is now with our production department. 

Kind regards, 

on behalf of

Dr. Theodoros M. Bampouras 

Academic Editor

PLOS ONE